# Categorical Schrödinger Bridge Matching

**Grigoriy Ksenofontov** [1] [2]  **Alexander Korotin** [1] [3]

## Abstract

The Schrödinger Bridge (SB) is a powerful framework for solving generative modeling tasks such as unpaired domain translation. Most SB-related research focuses on continuous data space $\mathbb{R}^D$ and leaves open theoretical and algorithmic questions about applying SB methods to discrete data, e.g, on finite spaces $\mathbb{S}^D$. Notable examples of such sets $\mathbb{S}$ are codebooks of vector-quantized (VQ) representations of modern autoencoders, tokens in texts, categories of atoms in molecules, etc. In this paper, we provide a theoretical and algorithmic foundation for solving SB in discrete spaces using the recently introduced Iterative Markovian Fitting (IMF) procedure. Specifically, we theoretically justify the convergence of discrete-time IMF (D-IMF) to SB in discrete spaces. This enables us to develop a practical computational algorithm for SB, which we call Categorical Schrödinger Bridge Matching (CSBM). We show the performance of CSBM via a series of experiments with synthetic data and VQ representations of images. The code of CSBM is available at this repository.

## 1 Introduction

The Schrödinger bridge (Schrödinger, 1931, SB) problem has recently attracted the attention of the machine learning community due to its relevance to modern challenges in generative modeling and unpaired learning. Recently, a variety of methods have been proposed to solve SB in *continuous spaces*; see (Gushchin et al., 2023b) for a recent survey.

One modern approach to solving SB is the Iterative Markovian Fitting (IMF) framework (Peluchetti, 2023; Shi et al., 2023; Gushchin et al., 2024b). Specifically, within this framework, the discrete-time IMF procedure (Gushchin

---

[1]Skoltech, Moscow, Russia [2]MIPT, Moscow, Russia [3]AIRI, Moscow, Russia. Correspondence to: Grigoriy Ksenofontov <g.ksenofontov@skoltech.ru>, Alexander Korotin <a.korotin@skoltech.ru>.

*Proceedings of the 42$^{nd}$ International Conference on Machine Learning*, Vancouver, Canada. PMLR 267, 2025. Copyright 2025 by the author(s).

et al., 2024b, D-IMF) has shown promising results in certain unpaired learning problems, enabling faster generation (inference) times than its predecessors.

Unfortunately, the D-IMF procedure heavily relies on certain theoretical properties of particular SB setups in continuous spaces. At the same time, a vast amount of real-world data is either *discrete by nature*, such as texts (Austin et al., 2021; Gat et al., 2024), molecular graphs (Vignac et al., 2022; Qin et al., 2024; Luo et al., 2024), sequences (Campbell et al., 2024), etc., or *discrete by construction* like vector-quantized representations of images and audio (Van Den Oord et al., 2017; Esser et al., 2021). These cases highlight a fundamental limitation, as D-IMF is not directly applicable to such data. In this work, we address this gap by making the following **contributions:**

- **Theory.** We provide the theoretical grounds for applying the D-IMF to solve the SB problem in discrete spaces.

- **Practice.** We provide a computational algorithm to implement the D-IMF in practice for discrete spaces.

**Notations.** Consider a *state space* $\mathcal{X}$ and a *time set* $\{t_n\}_{n=0}^{N+1}$, where $0 = t_0 < t_1 < \cdots < t_N < t_{N+1} = 1$ are $N \geq 1$ time moments. The space $\mathcal{X}^{N+2}$ is referred to as the *path space* and represents all possible trajectories $(x_0, x_{\text{in}}, x_{t_{N+1}})$, where $x_{\text{in}} \stackrel{\text{def}}{=} (x_{t_1}, \ldots, x_{t_N})$ corresponds to the intermediate states. Let $\mathcal{P}(\mathcal{X}^{N+2})$ be the space of probability distributions over paths. Each $q \in \mathcal{P}(\mathcal{X}^{N+2})$ can be interpreted as a discrete in time $\mathcal{X}$-valued stochastic process. We use $q(x_0, x_{\text{in}}, x_{t_{N+1}})$ to denote its density at $(x_0, x_{\text{in}}, x_{t_{N+1}}) \in \mathcal{X}^{N+2}$ and use $q(\cdot|\cdot)$ to denote its conditional distributions, e.g., $q(x_1|x_0)$, $q(x_{\text{in}}|x_0, x_1)$. Finally, we introduce $\mathcal{M}(\mathcal{X}^{N+2}) \subset \mathcal{P}(\mathcal{X}^{N+2})$ as the set of all *Markov processes* $q$, i.e., those processes which satisfy the equality $q(x_0, x_{\text{in}}, x_{t_{N+1}}) = q(x_0) \prod_{n=1}^{N+1} q(x_{t_n}|x_{t_{n-1}})$.

## 2 Background and Related Works

In this section, we review the formulation and existing approaches to the Schrödinger Bridge (SB) problem, with a focus on its generative applications. We begin with the static SB problem (§2.1). Next, we highlight the challenges of extending SB methods from continuous to discrete state spaces (§2.3). We proceed to the dynamic SB formulation,

motivating its importance in practice (§2.4). This leads to the Iterative Markovian Fitting (IMF) procedure and its discrete-time variant D-IMF (§2.5). Finally, we summarize the known characterizations of SB (Table 1) and identify the object of our study, namely, establishing theoretical guarantees for the discrete state and time setting (§2.6).

## 2.1 The Static Schrödinger Bridge Problem

Consider two distributions $p_0, p_1 \in \mathcal{P}(\mathcal{X})$ and all distributions $q \in \mathcal{P}(\mathcal{X}^2)$ whose marginal distributions are $p_0, p_1$, respectively. The set of such distributions $\Pi(p_0, p_1) \subset \mathcal{P}(\mathcal{X}^2)$ is called the set of *transport plans*. In addition, suppose we are given a reference distribution $q^{\text{ref}} \in \mathcal{P}(\mathcal{X}^2)$.

*The Static Schrödinger Bridge (SB) problem* (Schrödinger, 1931; Léonard, 2013) consists of finding the transport plan $q \in \Pi(p_0, p_1)$ closest to $q^{\text{ref}}$ in terms of the Kullback–Leibler (KL) divergence:

$$q^*(x_0, x_1) = \underset{q \in \Pi(p_0, p_1)}{\text{argmin}} \ \text{KL}(q(x_0, x_1) || q^{\text{ref}}(x_0, x_1)), \quad (1)$$

With mild assumptions on components of the problem $(\mathcal{X}, p_0, p_1, q^{\text{ref}})$, the solution $q^*$ to this problem uniquely exists; it is called the static SB.

Notably, the static SB problem is equivalent to another well-celebrated problem – the *Entropic Optimal Transport* (Cuturi, 2013, EOT). Indeed, (1) can be written as

$$\min_{q \in \Pi(p_0, p_1)} \mathbb{E}_{q(x_0, x_1)} \log \frac{q(x_0, x_1)}{q^{\text{ref}}(x_0, x_1)} =$$

$$\min_{q \in \Pi(p_0, p_1)} \left\{ \mathbb{E}_{q(x_0, x_1)} \underbrace{\left[ -\log q^{\text{ref}}(x_0, x_1) \right]}_{\overset{\text{def}}{=} c(x_0, x_1)} - H(q) \right\} =$$

$$\min_{q \in \Pi(p_0, p_1)} \left\{ \mathbb{E}_{q(x_0, x_1)} c(x_0, x_1) - H(q) \right\}, \quad (2)$$

where $H(q)$ denotes the entropy of transport plan $q(x_0, x_1)$ and $c(x_0, x_1)$ is a transport cost function.

## 2.2 Practical Learning Setup of SB

Over the last decade, researchers have approached SB/EOT problems in various studies because of their relevance to real-world tasks (Peyré et al., 2019; Gushchin et al., 2023b). In our paper, we consider the following learning setup, which is usually called the *generative* setup.

> We assume that a learner is given empirical datasets $\{x_0^m\}_{m=1}^M \subset \mathcal{X}$ and $\{x_1^k\}_{k=1}^K \subset \mathcal{X}$, which are i.i.d. samples from unknown data distributions $p_0$ and $p_1$, respectively. The goal is to leverage these samples to find a solution $\widehat{q} \approx q^*$ to the SB problem (2) between the distributions $p_0, p_1$. The solution should permit the **out-of-sample estimation**, i.e., for any $x_0^{\text{new}}$, one should be able to generate new $x_1^{\text{new}} \sim \widehat{q}(x_1 | x_0^{\text{new}})$.

In the related literature, this setup is mainly explored in the context of unpaired (unsupervised) domain translation. In this task, the datasets consist of samples from two different data distributions (domains), and the goal is to learn a transformation from one domain to the other (Zhu et al., 2017, Figure 2). The problem is inherently ill-posed because, theoretically, there may be multiple possible transformations. In many applications of unpaired learning, it is crucial to preserve semantic information during the translation, for example, the image content in image-to-image translation. Therefore, SB and EOT are suitable tools for this task as they allow controlling the properties of the learned translation by selecting the reference distribution $q^{\text{ref}}$ in (1) or the transport cost $c$ in (2). Over the last several years, many such SB/EOT methods for unpaired learning have been developed; see (Gushchin et al., 2023b) for a survey.

## 2.3 Discrete and Continuous State Space $\mathcal{X}$ in SB

Most methods (Mokrov et al., 2024; De Bortoli et al., 2021; Vargas et al., 2021; Gushchin et al., 2023a; 2024b; Korotin et al., 2024; Gushchin et al., 2024a; Shi et al., 2023; Liu et al., 2022a; Chen et al., 2022) use neural networks to approximate $q^*$ and *specifically* focus on solving SB in **continuous state spaces**, e.g., $\mathcal{X} = \mathbb{R}^D$. This allows us to apply SB to many unpaired translation problems, e.g., the above-mentioned image-to-image translation or biological tasks related to the analysis and modeling of single-cell data (Pariset et al., 2023; Tong et al., 2024).

Despite advances in computational SB methods, significant challenges remain when adapting these generative approaches to **discrete state spaces** $\mathcal{X}$:

1. Their underlying methodological principles are mostly incompatible with discrete spaces $\mathcal{X}$. For example, (Shi et al., 2023; Gushchin et al., 2023a; Vargas et al., 2021; Liu et al., 2022a) use stochastic differential equations (SDE) which are not straightforward to generalize and use in discrete spaces; (Mokrov et al., 2024) heavily relies on MCMC sampling from unnormalized density which is also a separate challenge for large discrete spaces $\mathcal{X}$; (Gushchin et al., 2024a; Korotin et al., 2024; Gushchin et al., 2024b) theoretically work only for the EOT problem with the quadratic cost on $\mathcal{X} = \mathbb{R}^D$, etc.

2. Extending any generative modeling techniques to discrete data is usually a challenge. For example, models such as GANs (Goodfellow et al., 2014) require back-propagation through the generator – for discrete data is usually done via heuristics related to the Gumbel trick (Jang et al., 2017); flow matching methods (Liu et al., 2022b) can be used for discrete data (Gat et al., 2024) but require numerous methodological changes, etc.

At the same time, a significant portion of modern data is

inherently discrete, as discussed in §1. Despite its prevalence, the Schrödinger Bridge framework for discrete spaces remains underdeveloped, motivating our focus.

> We assume that the state space $\mathcal{X}$ is discrete and represented as $\mathcal{X} = \mathbb{S}^D$. Here $\mathbb{S}$ is a finite set, and for convenience, we say that it is the space of categories, e.g., $\mathbb{S} = \{1, 2, \ldots, S\}$. One may also consider $\mathcal{X} = \mathbb{S}_1 \times \cdots \times \mathbb{S}_D$ for $D$ categorical sets. This does not make any principal difference, so we use $\mathbb{S}_1 = \cdots = \mathbb{S}_D$ to keep the paper's exposition simple.

**Discrete EOT Methods.** We would like to mention, for the sake of completeness, that there is a broad area of research known as discrete EOT, which might appear to be closely related to our work. It includes, e.g., the well-celebrated Sinkhorn algorithm (Cuturi, 2013) and gradient-based methods (Dvurechensky et al., 2018; Dvurechenskii et al., 2018). However, such algorithms **are not relevant** to our work, as they consider a different setting from the generative one (§2.2) and target different problems. Specifically, discrete EOT assumes that the available data samples are themselves discrete distributions, i.e., $p_0 = \frac{1}{M} \sum_{m=1}^{M} \delta_{x_0^m}$, $p_1 = \frac{1}{K} \sum_{k=1}^{K} \delta_{x_0^k}$ (the weights may be non-uniform), and the goal is to find a bi-stochastic matrix $\in \mathbb{R}^{M \times K}$ (a.k.a. the discrete EOT plan) which optimally matches the given samples. Since this matrix is a discrete object, such methods are called discrete. Works (Hütter & Rigollet, 2021; Pooladian & Niles-Weed, 2021; Manole et al., 2024; Deb et al., 2021) aim to advance discrete EOT methods to be used in generative setups by providing out-of-sample estimators. However, they work only for continuous state space $\mathcal{X} = \mathbb{R}^D$. It remains an open question whether discrete solvers can be used for generative scenarios in discrete space $\mathcal{X} = \mathbb{S}^D$.

### 2.4 From Static to Dynamic SB Problems

The static SB problem (1) can be thought of as a problem of finding a stochastic process acting at times $t = 0, 1$. Usually, one considers an extension of this problem by incorporating additional time moments (De Bortoli et al., 2021; Gushchin et al., 2024b). Let us introduce $N \geq 1$ intermediate time points $0 = t_0 < t_1 < \cdots < t_N < t_{N+1} = 1$, extending $q$ to these moments. Consequently, $q$ becomes a process over the states at all time steps, i.e., $q \in \mathcal{P}(\mathcal{X}^{N+2})$. Similarly to the static formulation (1), let us be given marginal distributions $p_0, p_1 \in \mathcal{P}(\mathcal{X})$ with a reference process $q^{\text{ref}} \in \mathcal{P}(\mathcal{X}^{N+2})$. Then the *dynamic Schrödinger Bridge* problem is

$$\min_{q \in \Pi_N(p_0, p_1)} \mathrm{KL}(q(x_0, x_{\text{in}}, x_1) || q^{\text{ref}}(x_0, x_{\text{in}}, x_1)), \quad (3)$$

where $\Pi_N(p_0, p_1) \subset \mathcal{P}(\mathcal{X}^{N+2})$ is a set of all discrete-time stochastic processes in which initial and terminal marginal

distributions are $p_0$ and $p_1$. In turn, the solution $q^*$ to this itself becomes an $\mathcal{X}$-valued stochastic process. Note that:

$$\mathrm{KL}(q(x_0, x_{\text{in}}, x_1) || q^{\text{ref}}(x_0, x_{\text{in}}, x_1)) =$$
$$\mathrm{KL}(q(x_0, x_1) || q^{\text{ref}}(x_0, x_1)) +$$
$$\mathbb{E}_{q(x_0, x_1)} \left[ \mathrm{KL}(q(x_{\text{in}}|x_0, x_1) || q^{\text{ref}}(x_{\text{in}}|x_0, x_1)) \right]. \quad (4)$$

Since conditional distributions $q(x_{\text{in}}|x_0, x_1)$ can be chosen independently of $q(x_0, x_1)$, we can consider $q(x_{\text{in}}|x_0, x_1) = q^{\text{ref}}(x_{\text{in}}|x_0, x_1)$. It follows that the second term becomes 0 for every $x_0, x_1$. As a result, we see that the joint distribution $q^*(x_0, x_1)$ for time $t = 0, 1$ of the dynamic SB (3) is the solution to the static SB (1) for the reference distribution given by the $q^{\text{ref}}(x_0, x_1)$.

At this point, a reader may naturally wonder: *why does one consider the more complicated Dynamic SB, especially considering that it boils down to simpler Static SB?*

In short, the dynamic solution allows for leveraging the so-called reciprocal and Markov properties of $q^*$ (it is discussed below), which can be effectively utilized in developing computational algorithms for SB (Liu et al., 2023; Shi et al., 2023; Peluchetti, 2023). In fact, **most** of the computational methods listed at the beginning of §2.3 operate with the dynamic SB formulation. While some methods (De Bortoli et al., 2021; Gushchin et al., 2024b) consider formulation (3) with discrete time and finite amount $N$ of time moments, (Shi et al., 2023; Chen et al., 2022; Gushchin et al., 2024a) work with continuous time $t \in [0, 1]$. **Informally**, one may identify it with discrete time but $N = \infty$. In discussions, we will refer to the continuous time case this way in the rest of the paper *to avoid unnecessary objects and notations*.

> The scope of our paper is exclusively the discrete-time in dynamic SB ($N < \infty$) as it is more transparent and feasible to analyze.

To conclude this section, we introduce an important definition that is specifically relevant to the dynamic SB.

**Reciprocal Processes.** A process $r \in \mathcal{P}(\mathcal{X}^{N+2})$ is called a reciprocal process with respect to the reference process $q^{\text{ref}}$ if its conditional distributions given the endpoints $x_0, x_1$ match those of the reference process, i.e.:

$$r(x_{\text{in}} \mid x_0, x_1) = q^{\text{ref}}(x_{\text{in}} \mid x_0, x_1).$$

The set of all reciprocal processes for the reference process $q^{\text{ref}}$ is denoted by $\mathcal{R}^{\text{ref}}(\mathcal{X}^{N+2}) \subset \mathcal{P}(\mathcal{X}^{N+2})$.

### 2.5 Iterative Markovian Fitting (IMF) Procedure

In practice, the most commonly considered case of dynamic SB is when $q^{\text{ref}} \in \mathcal{M}(\mathcal{X}^{N+2}) \subset \mathcal{P}(\mathcal{X}^{N+2})$, i.e., $q^{\text{ref}}$ is a

| | Continuous time $(N = \infty)$ | | Discrete time $(N < \infty)$ | |
|---|---|---|---|---|
| | *Theory* (SB characterization) | *Practice* (SB algorithm) | *Theory* (SB characterization) | *Practice* (SB algorithm) |
| **Continuous space** $\mathcal{X} = \mathbb{R}^D$ | Theorem 3.2 (Léonard et al., 2014) | DSBM §4 (Shi et al., 2023) | Theorem 3.1 (Gushchin et al., 2024b) | ASBM §3.5 (Gushchin et al., 2024b) |
| **Discrete space** $\mathcal{X} = \mathbb{S}^D$ | | DDSBM §3.1 (Kim et al., 2024) | **Our work** (§3) | |

*Table 1.* A summary of SB problem setups and existing (D-)IMF-related results. The table lists theoretical statements characterizing the SB solution (*as the unique both Markovian and reciprocal process between two given distributions*) which allows to apply the (D-)IMF procedure to provably get the SB solution $q^*$, see (Shi et al., 2023, Theorem 8). The table also lists related computational algorithms.

*Markovian process.* In this case, the solution $q^*$ to SB is also known to be a Markovian process. This feature motivated the researchers to develop the *Iterative Markovian Fitting* (IMF) procedure for solving SB based on Markovian and reciprocal projections of stochastic processes.

Originally, the IMF procedure (Peluchetti, 2023; Shi et al., 2023) was considered the continuous time ($N = \infty$), but recently, it has been extended to the finite amount of time moments (Gushchin et al., 2024b), i.e., $N < \infty$. We recall their definitions of the projections for finite $N$. In this case, the procedure is called the **D-IMF** (discrete-time IMF).

**Reciprocal Projection.** Consider a process $q \in \mathcal{P}(\mathcal{X}^{N+2})$. Then the reciprocal projection $proj_{\mathcal{R}^{\text{ref}}}(q)$ with respect to the reference process $q^{\text{ref}}$ is a process given by:

$$[proj_{\mathcal{R}^{\text{ref}}}(q)](x_0, x_{\text{in}}, x_1) = q^{\text{ref}}(x_{\text{in}}|x_0, x_1)q(x_0, x_1).$$

**Markovian Projection.** Consider $q \in \mathcal{P}(\mathcal{X}^{N+2})$. Then the Markovian projection $proj_{\mathcal{M}}(q)$ is given by:

$$[proj_{\mathcal{M}}(q)](x_0, x_{\text{in}}, x_1) =$$
$$= \underbrace{q(x_0) \prod_{n=1}^{N+1} q(x_{t_n}|x_{t_{n-1}})}_{\text{forward representation}} =$$
$$= \underbrace{q(x_1) \prod_{n=1}^{N+1} q(x_{t_{n-1}}|x_{t_n})}_{\text{backward representation}} \quad (5)$$

The reciprocal projection obviously preserves the joint distribution $q(x_0, x_1)$ of a process at time moments $t = 0, 1$. The Markovian projection, in general, alters $q(x_0, x_1)$ but preserves the joint distributions $\{q(x_{t_n}, x_{t_{n-1}})\}_{n=1}^{N+1}$ at neighboring time moments and the marginal distributions $q(x_{t_n})$.

**The D-IMF procedure** is initialized with any process $q^0 \in \Pi_N(p_0, p_1)$. Then the procedure alternates between

reciprocal $proj_{\mathcal{R}^{\text{ref}}}$ and Markovian $proj_{\mathcal{M}}$ projections:

$$q^{2l+1} = proj_{\mathcal{R}^{\text{ref}}}(q^{2l}),$$
$$q^{2l+2} = proj_{\mathcal{M}}(q^{2l+1}). \quad (6)$$

Since both the Markovian and reciprocal projections preserve marginals $p_0, p_1$ at times $t = 0, 1$, respectively, we have that each $q^l \in \Pi_N(p_0, p_1)$. In certain configurations of $N, \mathcal{X}, q^{\text{ref}}$, IMF provably converges to the dynamic SB $q^*$ in KL, i.e., $\lim_{l \to \infty} \text{KL}(q^l \| q^*) = 0$. Specifically, the convergence easily follows from the generic proof argument in (Shi et al., 2023, Theorem 8) *as soon as it is known that $q^*$ is the unique process in $\Pi_N(p_0, p_1)$ that is both Markovian and reciprocal.* We provide Table 1, summarizing the configurations for which this **characterization** of SB is known. We also list the related practical algorithms.

Finally, we would like to emphasize that the *convergence rate of the (D-)IMF procedure notably depends on the number $N$ of time steps.* In fact, for each $N$ it is its own separate procedure with a different Markovian projection (5), see (Gushchin et al., 2024b, Figure 6a).

### 2.6 Object of Study

As it is clear from Table 1, for the setup with the discrete space $\mathcal{X} = \mathbb{S}^D$ and finite amount of time moments $N < \infty$, there is still no theoretical guarantee that the SB is the unique Markovian and reciprocal process. This leaves a large gap in D-IMF usage in this case, and we close it in our paper.

At the same time, we note that there is a very recent IMF-based algorithm DDSBM (Kim et al., 2024) for the discrete state space $\mathcal{X}$ but continuous time ($N = \infty$). However, since working with continuous time is infeasible in practice, the authors discretize the time grid to a large finite $N$. Due to this, the authors apply the D-IMF procedure, although it still lacks any theoretical ground in this case. In contrast, our work shows that *theoretically even $N = 1$ is enough.*

# 3 Categorical Schrödinger Bridge Matching

We start by establishing the convergence of the D-IMF framework ($N < \infty$) to the SB under a general Markov reference process (§3.1) with the proofs in Appendix B. Then we provide a practical optimization procedure and implementation details of the proposed method (§3.2).

## 3.1 Theoretical Foundation

The result of (Gushchin et al., 2024b, Theorem 3.6) characterizes the SB solution in $\mathcal{X} = \mathbb{R}^D$ and $N < \infty$ as the unique Markovian and Reciprocal process which allows the usage of D-IMF procedure. However, that proof assumes a specific reference process $q^{\text{ref}} = q^W$ induced by the Wiener process $W$ (EOT with the quadratic cost) and thus cannot handle a general Markov $q^{\text{ref}}$ or discrete $\mathcal{X}$.

Below we provide our main theoretical result for the *discrete* space $\mathcal{X}$ and *general* Markov reference process $q^{\text{ref}}$ which characterizes SB and immediately allows the usage of D-IMF ($N < \infty$) procedure to get it.[1]

> **Theorem 3.1** (Characterization of the solution for the dynamic SB problem on a discrete space $\mathcal{X}$ with a Markovian reference $q^{\text{ref}}$). *Let $\mathcal{X}$ be a finite discrete space and let $p_0, p_1 \in \mathcal{P}(\mathcal{X})$ be distributions with full support. Let $q^{\text{ref}} \in \mathcal{M}(\mathcal{X}^{N+2})$ be a reference Markov process with full support on $\mathcal{X}^{N+2}$. If $q^* \in \mathcal{P}(\mathcal{X}^{N+2})$ satisfies the following conditions:*
>
> 1. *$q^*(x_0) = p_0(x_0)$ and $q^*(x_1) = p_1(x_1)$, i.e., $q^*(x_0, x_1)$ is a **transport plan** from $\Pi(p_0, p_1)$;*
>
> 2. *$q^* \in \mathcal{M}(\mathcal{X}^{N+2})$ and $q^* \in \mathcal{R}^{\text{ref}}(\mathcal{X}^{N+2})$, i.e., $q^*$ is both the **reciprocal** and **Markov**,*
>
> *then $q^*$ is the unique solution of the dynamic SB (3).*

Our theorem immediately yields the following corollary.

**Corollary 3.2** (Convergence of D-IMF on discrete spaces). *The sequence $\{q^l\}_{l=0}^{\infty}$ produced by the D-IMF procedure on a discrete space $\mathcal{X}$ and for a Markov reference process from the theorem above converges to $q^*$ in KL:*

$$\lim_{l \to \infty} KL\left(q^l \| q^*\right) = 0.$$

## 3.2 Practical Implementation

In this subsection, we discuss our computational algorithm to implement D-IMF and get the SB problem solution $q^*$.

Since we consider a finite amount $N$ of time steps, the processes $q \in \mathcal{P}(\mathcal{X}^{N+2})$ are discrete-time Markov chains

---

[1]In fact, our proof argument can be applied to any $\mathcal{X}$, i.e., not only discrete, thus, the ASBM algorithm (Gushchin et al., 2024b) for *continuous* $\mathcal{X} = \mathbb{R}^D$ can be applied for general Markov $q^{\text{ref}}$.

(DTMC). A DTMC is defined by $N + 1$ transition matrices $Q_n$ of size $|\mathcal{X}| \times |\mathcal{X}|$, where $[Q_n]_{x_{t_{n-1}} x_{t_n}}$ represents the probability of transitioning from state $x_{t_{n-1}}$ to state $x_{t_n}$:

$$q(x_{t_n} | x_{t_{n-1}}) = [Q_n]_{x_{t_{n-1}} x_{t_n}}.$$

Thus, in theory, one can model any such DTMC $q$ explicitly. However, in practice, the size $|\mathcal{X}|$ may be large. In particular, we consider the case $\mathcal{X} = \mathbb{S}^D$, where $\mathbb{S}$ is a categorical space leading to exponential amount $S^D$ of elements in $\mathcal{X}$.

This raises two natural questions: **(a)** how to choose a reference process $q^{\text{ref}}$ and work with it? and **(b)** how to parameterize and update the process $q$ during D-IMF steps? Both these questions will be answered in the following generic discussion about the parameterization and implementation of reciprocal and Markovian projections.

**3.2.1 Implementing the Reciprocal Projection.** The reciprocal projection is rather straightforward if we can draw samples from our current process $q(x_0, x_1)$ and the reference bridge $q^{\text{ref}}(x_{t_{n-1}} | x_0, x_1)$. Indeed, sampling $(x_0, x_{t_{n-1}}, x_1) \sim proj_{\mathcal{R}^{\text{ref}}}(q)$ is just merging these two.

**3.2.2 Choosing a Reference Process.** As it is clear from the paragraph above, it is reasonable to consider reference processes $q^{\text{ref}} \in \mathcal{M}(\mathcal{X}^{N+2})$ for which sampling from their bridge $q^{\text{ref}}(x_{t_{n-1}} | x_0, x_1)$ is easy. We give two popular examples of $q^{\text{ref}}$ which appear in related work (Austin et al., 2021) that lead to practically meaningful cost $c$ for EOT (2). For both examples, we start with dimension $D = 1$.

**Case 1 (Uniform Reference $q^{\text{unif}}$).** In this case, we assume that the set of categories $\mathbb{S}$ is unordered, e.g., atom types, text tokens, latent variables, etc. Define a process where the state stays in the current category $x_{t_{n-1}}$ with high probability, while the remaining probability is distributed uniformly among all other categories. This process $q^{\text{unif}}$ is called *uniform* and has transitions matrices $Q_n$:

$$[Q_n]_{x_{t_{n-1}} x_{t_n}} = \begin{cases} 1 - \alpha, & \text{if } x_{t_n} = x_{t_{n-1}}, \\ \frac{\alpha}{S-1}, & \text{if } x_{t_n} \neq x_{t_{n-1}}, \end{cases} \quad (7)$$

where $\alpha \in [0, 1]$ is the *stochasticity parameter* that controls the probability of transitioning to a different category.

**Case 2 (Gaussian Reference $q^{\text{gauss}}$).** If we know that the categories are ordered, specifically, $\mathbb{S} = (1, 2, \ldots, S)$, and two neighboring categories are assumed to be related, the transitions may be chosen to reflect this. Consider the *Gaussian*-like reference process $q^{\text{gauss}}$ with $[Q_n]_{x_{t_{n-1}} x_{t_n}} =$

$$\begin{cases} \frac{\exp\left(-\frac{4(x_{t_n} - x_{t_{n-1}})^2}{(\alpha \Delta)^2}\right)}{\sum_{\delta=-\Delta}^{\Delta} \exp\left(-\frac{4\delta^2}{(\alpha \Delta)^2}\right)}, & x_{t_n} \neq x_{t_{n-1}}, \\ 1 - \sum_{x_{t_n} \neq x_{t_{n-1}}} [Q_n]_{x_{t_{n-1}} x_{t_n}}, & x_{t_n} = x_{t_{n-1}}, \end{cases} \quad (8)$$

where $\alpha > 0$ is an analog of the variance parameter, and $\Delta = S - 1$ is a maximum distance between categories.

**Dimension $D > 1$.** The construction of $q^{\text{unif}}$ (or $q^{\text{gauss}}$) generalizes to higher $D$ by combining several such independent processes (one per dimension). The bridges $q^{\text{ref}}(x_{\text{in}}|x_0, x_1)$ can be easily derived analytically and sampled thanks to the Markov property and the Bayes' formula.

For more details on the construction and selection of reference processes $q^{\text{ref}}$, please refer to Appendix D.1.

### 3.2.3 Parameterization of the Learnable Process.

There are $|\mathbb{S}^D| = S^D$ possible states $x = (x^1, \ldots, x^D)$ in the space, where $S$ is the number of categories for each variable. Consequently, each transition matrix $Q_n$ is of size $S^D \times S^D$, i.e., it grows exponentially in dimension $D$. Due to this, explicit modeling of the transition matrices of the process that we learn is computationally infeasible. We follow the standard practice in discrete generative models (Hoogeboom et al., 2021; Austin et al., 2021; Gat et al., 2024; Campbell et al., 2024) and model the transition probability via combining two popular techniques: posterior sampling and factorization over the dimensions. Firstly, we parameterize the transitions $q_\theta(x_{t_n}|x_{t_{n-1}})$ as follows:

$$q_\theta(x_{t_n}|x_{t_{n-1}}) = \mathbb{E}_{\widetilde{q}_\theta(\widetilde{x}_1|x_{t_{n-1}})}\left[q^{\text{ref}}(x_{t_n}|x_{t_{n-1}}, \widetilde{x}_1)\right], \quad (9)$$

where $\widetilde{q}_\theta(\widetilde{x}_1|x_{t_{n-1}})$ is a learnable distribution. This parameterization assumes that sampling of $x_{t_n}$ given $x_{t_{n-1}}$ can be done by first sampling some "endpoint" $\widetilde{x}_1 \sim \widetilde{q}_\theta(\widetilde{x}_1|x_{t_{n-1}})$, and then sampling from the bridge $q^{\text{ref}}(x_{t_n}|x_{t_{n-1}}, \widetilde{x}_1)$. Second, the parameterization for $\widetilde{q}_\theta(\widetilde{x}_1|x_{t_{n-1}})$ is factorized:

$$\widetilde{q}_\theta(\widetilde{x}_1|x_{t_{n-1}}) \approx \prod_{d=1}^{D} \widetilde{q}_\theta(\widetilde{x}_1^d|x_{t_{n-1}}).$$

In this case, for each $x_{t_{n-1}}$, we just need to predict a row-stochastic $D \times S$ matrix of probabilities $\widetilde{q}_\theta(\widetilde{x}_1^d|x_{t_{n-1}})$. See Appendix A for a discussion of the limitations of this approach. Following the common practices, we employ a neural network $S^D \to D \times S$ which outputs a row-stochastic matrix for each input $x_{t_{n-1}}$. Typically, predicting endpoints at each time step $n-1$ would require $N+1$ distinct models for each $\widetilde{q}_\theta(\widetilde{x}_1|x_{t_{n-1}})$. Instead, we use a single neural network with an additional input indicating the timestep.

### 3.2.4 Implementing the Markovian Projection.

The Markovian projection is a little bit more complex than the reciprocal one and requires learning a process. From §2.5, the goal of the projection is to find a Markov process whose transition probabilities match those of the given reciprocal process $q$. Fortunately, we show that this can be achieved by minimizing an objective that closely resembles the optimization of the variational bound used in diffusion models (Ho et al., 2020; Austin et al., 2021; Hoogeboom et al., 2021).

**Proposition 3.3.** *Let $q \in \mathcal{R}^{\text{ref}}(\mathcal{X}^{N+2})$ be a given reciprocal process. Then, the Markovian projection $proj_\mathcal{M}(q) \in \mathcal{M}(\mathcal{X}^{N+2})$ can be obtained by minimizing:*

---

**Algorithm 1** Categorical SB matching (CSBM)

---

**Input:** number of intermediate time steps $N$;
number of outer iterations $L \in \mathbb{N}$;
initial coupling $q^0(x_0, x_1)$;
reference process $q^{\text{ref}}$.
**Output:** forward model $q_\theta(x_{t_n}|x_{t_{n-1}})$;
backward model $q_\eta(x_{t_{n-1}}|x_{t_n})$.
  **for** $l = 1$ **to** $L$ **do**
    **Forward step** (repeat until convergence)**:**
      Sample $n \sim U[1, N+1]$;
      Sample $(x_0, x_1) \sim p_1(x_1)\prod_{n=1}^{N+1} q_\eta(x_{t_{n-1}}|x_{t_n})$;
      Sample $x_{t_{n-1}} \sim q^{\text{ref}}(x_{t_{n-1}}|x_0, x_1)$;
      Train $q_\theta$ by minimizing $L_\theta$ (21);
    **Backward step** (repeat until convergence)**:**
      Sample $n \sim U[1, N+1]$;
      Sample $(x_0, x_1) \sim p_0(x_0)\prod_{n=1}^{N+1} q_\theta(x_{t_n}|x_{t_{n-1}})$;
      Sample $x_{t_n} \sim q^{\text{ref}}(x_{t_n}|x_0, x_1)$;
      Train $q_\eta$ by minimizing $L_\eta$ (22);
  **end for**

---

$$
\begin{aligned}
L(m) &\stackrel{\text{def}}{=} \mathbb{E}_{q(x_0, x_1)}\Bigg[\sum_{n=1}^{N} \mathbb{E}_{q^{\text{ref}}(x_{t_{n-1}}|x_0, x_1)} \\
&\text{KL}\left(q^{\text{ref}}(x_{t_n}|x_{t_{n-1}}, x_1)||m(x_{t_n}|x_{t_{n-1}})\right) - \\
&- \mathbb{E}_{q^{\text{ref}}(x_{t_N}|x_0, x_1)}\left[\log m(x_1|x_{t_N})\right]\Bigg], \quad (10)
\end{aligned}
$$

*among the Markov processes $m \in \mathcal{M}(\mathcal{X}^{N+2})$. Furthermore, this objective is also equivalent to optimizing $\sum_{n=1}^{N+1} \mathbb{E}_{q(x_{t_{n-1}})} KL\left(q(x_{t_n}|x_{t_{n-1}})||m(x_{t_n}|x_{t_{n-1}})\right)$.*

Note that the key distinction from standard losses in diffusion models, such as (Austin et al., 2021, Equation 1), lies in the sampling of $x_{t_{n-1}}$. Instead of drawing from the noising process $q^{\text{ref}}(x_{t_{n-1}}|x_1)$, it is sampled from the reference bridge distribution $q^{\text{ref}}(x_{t_{n-1}}|x_0, x_1)$. As a result, with the proposed parametrization and Markovian projection representation, we can effectively apply the learning methodology from D3PM (Austin et al., 2021). The explicit loss formulation is provided in Appendix D.2.

### 3.2.5 Practical Implementation of the D-IMF Procedure.

With the reciprocal and Markovian projections fully established, we now proceed to the implementation of the D-IMF procedure. This method is conventionally applied in a bidirectional manner (Shi et al., 2023; Gushchin et al., 2024b), incorporating both forward and backward representations (5). This is because training in a unidirectional manner has been shown to introduce an error in IMF (De Bortoli et al., 2024, Appendix I). Therefore, we follow a bidirectional approach, which naturally leads to the **Categorical Schrödinger Bridge Matching (CSBM)** Algorithm 1.

# 4 Experimental Illustrations

We evaluate our CSBM algorithm across several setups. First, we analyze the convergence of D-IMF on discrete data (§4.1). Then, we demonstrate how CSBM performs with different reference processes in 2D experiments (§4.2). Next, we test CSBM's ability to translate images using the colored MNIST dataset (§4.3), varying the number of steps $N$. We then present an experiment on the CelebA dataset (§4.4), showcasing CSBM's performance in a latent space. Finally, we explore the text domain by solving sentiment transfer on the Amazon Reviews dataset (Appendix C.4). Experimental details are provided in Appendix D.3 and additional immages in Appendix D.4.

## 4.1 Convergence of D-IMF on Discrete Spaces

In this section, we derive analytical expressions for D-IMF and compare its convergence on discrete data under several setups. As noted in §2.5, the Markovian projection preserves the one-step transition probabilities of the given process $q^{2l+1}$. Thus, our task reduces to replicating:

$$q^{2l+2}\big(x_{t_n}|x_{t_{n-1}}\big) = q^{2l+1}\big(x_{t_n}|x_{t_{n-1}}\big), \quad \forall n \in [1, N+1].$$

For each D-IMF iteration, these transition matrices can be extracted from the joint distribution:

$$q^{2l+1}(x_{t_n}, x_{t_{n-1}}) = \sum_{x_0, x_1 \in \mathcal{X}} \Big[ q^{2l+1}(x_0, x_1) \cdot$$
$$\cdot \, q^{\text{ref}}\big(x_{t_n}|x_0, x_1\big) q^{\text{ref}}\big(x_{t_{n-1}}|x_{t_n}, x_1\big) \Big],$$

where $q^{\text{ref}}\big(x_{t_n}|x_0, x_1\big)$ and $q^{\text{ref}}\big(x_{t_{n-1}}|x_{t_n}, x_1\big)$ could be derived using Markov property and Bayes' formula.

Given $q^{2l+1}(x_{t_n}, x_{t_{n-1}})$, we obtain the desired transition distribution $q^{2l+2}(x_{t_n}|x_{t_{n-1}}) = [Q_n^{2l+2}]_{x_{t_{n-1}}, x_{t_n}}$ by normalizing the joint distribution over the marginal $q^{2l+1}(x_{t_{n-1}})$, which is computed by summing over all $x_{t_n} \in \mathbb{S}^D$ in $q^{2l+1}(x_{t_n}, x_{t_{n-1}})$. We then get the conditional distribution $q^{2l+2}(x_1|x_0)$ by multiplying the transition matrices $Q_n^{2l+2}$, i.e., $q^{2l+2}(x_1|x_0) = \big[\prod_{n=1}^{N+1} Q_n^{2l+2}\big]_{x_0, x_1}$.

Finally, we reweight this conditional distribution with $p_0(x_0)$ to obtain a new coupling $q^{2l+2}(x_0, x_1) = p_0(x_0)\big[\prod_{n=1}^{N+1} Q_n^{2l+2}\big]_{x_0, x_1}$ of the next iteration.

All of these equations are tractable and can be efficiently computed for small values of $S$ and $D$. Therefore, in our experiment, we solve the SB problem with $S = 50$ and $D = 1$ between the following marginals:

$$p_0(x_0) = \frac{1}{S}, \quad p_1(x_1) = \frac{x_1}{\sum_{s=1}^{S} s}.$$

To assess convergence as in Corollary 3.2, we also required to have the ground-truth bridge $q^*$, which we compute via

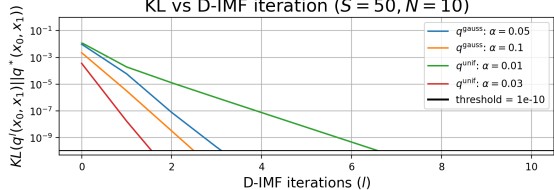

(a) Dependence on the stochastisity parameter $\alpha$.

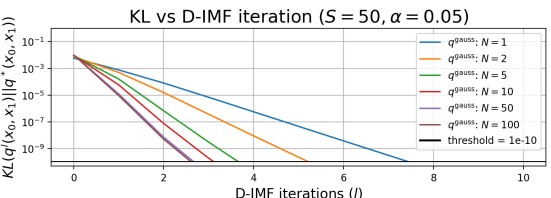

(b) Dependence on the number of time steps $N$ with $q^{\text{gauss}}$.

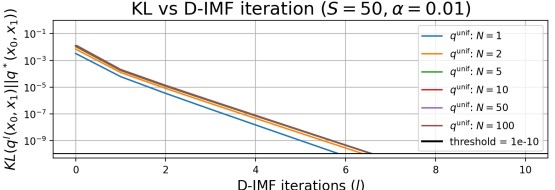

(c) Dependence on the number of time steps $N$ with $q^{\text{unif}}$.

*Figure 1.* Dependence of convergence of D-IMF procedure on discrete data under different $N$, $\alpha$ and $q^{\text{ref}}$.

the Sinkhorn algorithm (Cuturi, 2013). As a cost matrix, we use the negative logarithm of a cumulative transition matrix $\prod_{n=1}^{N+1} Q_n$. The resulting convergence curves, shown in Figure 1, indicate notably fast convergence of KL $\big(q^l \| q^*\big)$.

## 4.2 Illustrative 2D Experiments

In this experiment, we take the initial distribution $p_0$ as a 2D Gaussian and the target distribution $p_1$ as a Swiss Roll. Both are discretized into $S = 50$ categories, resulting in a 2-dimensional categorical space with $|\mathcal{X}| = S^2 = 50 \times 50$ points. Compared to the previous experiment, this setup involves working with $N$ matrices of size $2\,500 \times 2\,500$, making it a significantly more demanding computational task. Therefore, from now on, we solve the SB problem using our proposed Algorithm 1. The goal of this experiment is to examine the impact of the reference processes $q^{\text{gauss}}$ and $q^{\text{unif}}$. Thus, we train CSBM with $N = 10$ intermediate steps with different $\alpha$ and $q^{\text{ref}}$. For $q^{\text{gauss}}$, we test $\alpha \in \{0.02, 0.05\}$. In the case of $q^{\text{unif}}$ we use $\alpha \in \{0.01, 0.005\}$.

Figure 2 demonstrates that increasing the parameter $\alpha$ increases the number of jumps. In the case of $q^{\text{gauss}}$, the jumps mostly happen only to neighboring categories (Figures 2c and 2d). In the case of $q^{\text{unif}}$, the jumps happen to all categories (Figures 2e and 2f). This is aligned with the construction of the reference processes.

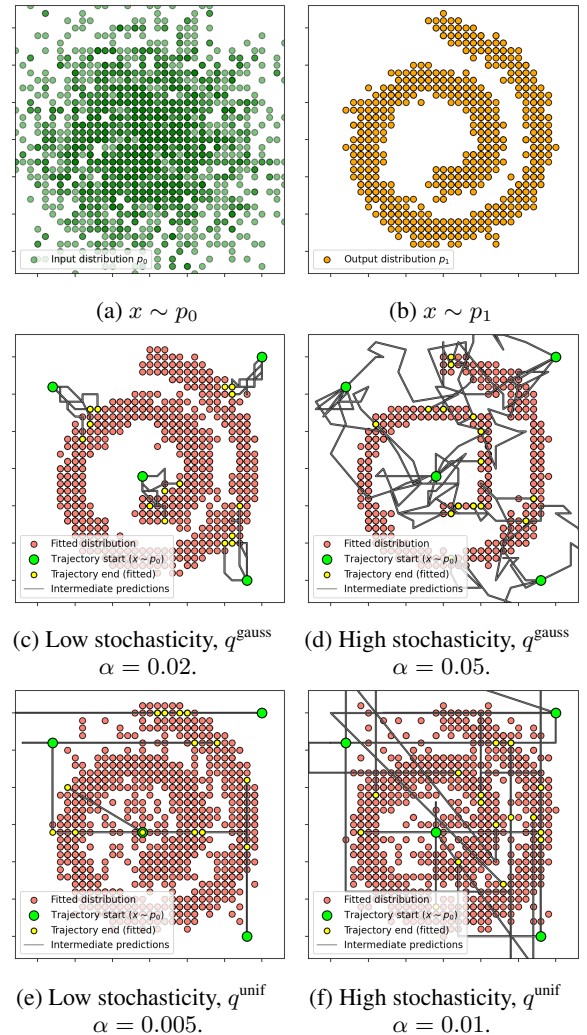

(a) $x \sim p_0$

(b) $x \sim p_1$

(c) Low stochasticity, $q^{\text{gauss}}$
$\alpha = 0.02$.

(d) High stochasticity, $q^{\text{gauss}}$
$\alpha = 0.05$.

(e) Low stochasticity, $q^{\text{unif}}$
$\alpha = 0.005$.

(f) High stochasticity, $q^{\text{unif}}$
$\alpha = 0.01$.

*Figure 2.* SB between 2D *Gaussian* and *Swiss-Roll* distributions learned by our CSBM algorithm with different reference processes $q^{\text{unif}}$ and $q^{\text{gauss}}$ with varying parameters $\alpha$.

**Remark.** Beyond the theoretical objectives established in Proposition 3.3, one can match the distributions using alternative loss functions, such as MSE, or through adversarial methods, as in ASBM (Gushchin et al., 2024b). For completeness, we conducted additional experiments using the MSE loss and observed results comparable to those obtained with KL. Details on the experimental setup and loss generalization are provided in Appendix C.1.

### 4.3 Unpaired Translation on Colored MNIST

Here, we work with the MNIST dataset with randomly colored digits. Inspired by (Gushchin et al., 2024b, Appendix C.3), we consider an unpaired translation problem between classes "2" and "3" of digits. In our case, we work in the discrete space of images, but not in a continuous space.

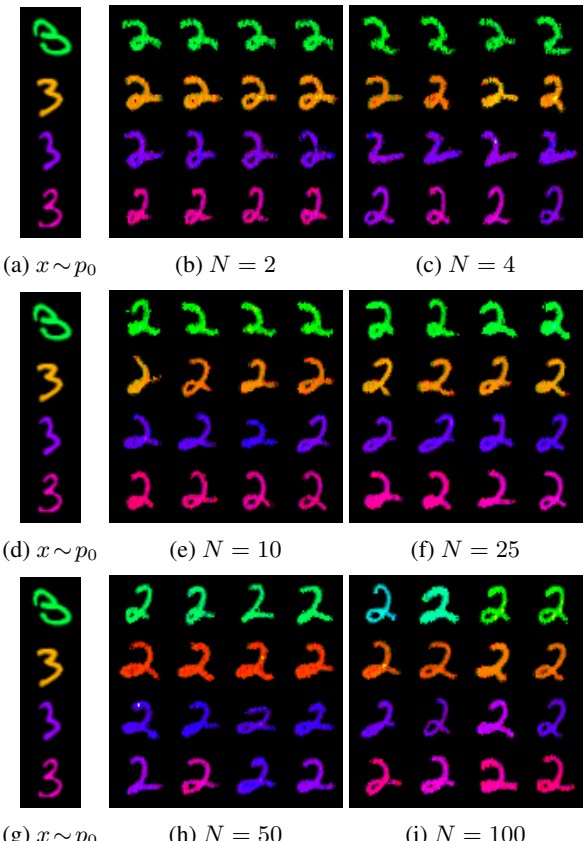

(a) $x \sim p_0$

(b) $N = 2$

(c) $N = 4$

(d) $x \sim p_0$

(e) $N = 10$

(f) $N = 25$

(g) $x \sim p_0$

(h) $N = 50$

(i) $N = 100$

*Figure 3.* Results of colored digits unpaired translation "3" → "2" learned by our CSBM algorithm with reference process $q^{\text{gauss}}$ and varying number of time moments $N$.

Specifically, each pixel is represented using three 8-bit channels (RGB), i.e., $S = 256$, and the data space is of size $256^D$, where $D = 32 \times 32 \times 3$. The goal of this experiment is to evaluate the capability of CSBM to perform unpaired translation with different numbers of intermediate steps $N$. Since each color channel values have an inherent order, we utilize the Gaussian reference process $q^{\text{gauss}}$ with $\alpha = 0.01$.

The results in Figure 3 suggest that even with a low $N = 2$, the generated outputs maintain decent visual quality and preserve the color. However, some pixelation appears in the samples, which is likely due to the factorization of the learned process (recall §3.2.3). The effect declines slightly as $N$ increases, reflecting a trade-off between model simplicity and the ability to capture inter-feature dependencies. Moreover, it can be observed that similarity reduces proportionally to $N$. We hypothesize that this issue is related to underfitting, since all models were trained with the same number of gradient updates. Presumably, a larger $N$ requires proportionally more updates to adequately train all transition probabilities (9). Additionally, we experiment with $q^{\text{unif}}$ with details provided in Appendix C.2.

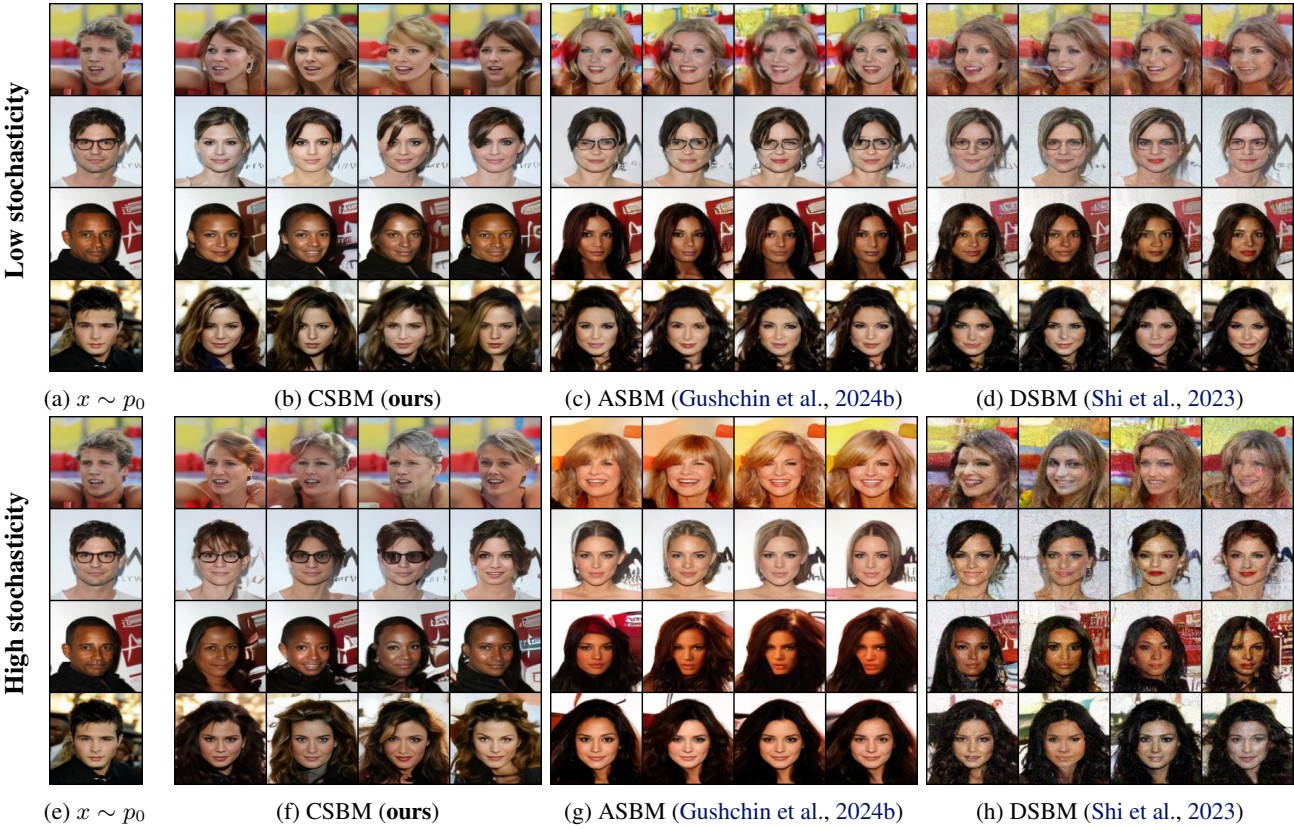

*Figure 4.* Comparison of *male → female* translation on the CelebA 128 × 128 dataset using CSBM (ours), ASBM, and DSBM. ASBM and DSBM operate in continuous pixel space, whereas CSBM operates in a discrete latent space of VQ-GAN (Esser et al., 2021). The low-stochasticity setting for CSBM corresponds to $\alpha = 0.005$, while the high-stochasticity setting corresponds to $\alpha = 0.01$ of the reference process $q^{\text{unif}}$. The images for ASBM and DSBM are taken from (Gushchin et al., 2024b).

## 4.4 Unpaired Translation of CelebA Faces

Here, we present an unpaired image-to-image translation experiment on the CelebA dataset using vector quantization. Specifically, we focus on translating images from the *male* to the *female* domain. We train VQ-GAN autoencoder (Esser et al., 2021) to represent 128 × 128 images as $D = 256$ features with $S = 1024$ categories (a.k.a. the codebook). This formulation reduces complexity, as the data to be modeled has a dimensionality of $S^D = 1024^{256}$. Indeed, this is smaller than the raw colored MNIST image space (§4.3) and considerably smaller than the raw pixel space of CelebA. As there is no clear relation between the elements of the codebook, we use uniform reference $q^{\text{ref}}$. We test $\alpha \in \{0.005, 0.01\}$ and $N = 100$.

For completeness, we compare our CSBM method with ASBM (Gushchin et al., 2024b) and DSBM (Shi et al., 2023), which operate in the continuous pixel space. For the rationale behind not training them in the latent space, see Appendix C.3. We take their results from (Gushchin et al., 2024b, §4.2). Qualitatively, we achieve comparable visual results (Figure 4). Notably, the background remains nearly

identical across all images for CSBM, which is not the case for all other methods, especially in high stochasticity setups.

*Table 2.* Metrics comparison of CSBM (**ours**), (Gushchin et al., 2024b, ASBM), and (Shi et al., 2023, DSBM) for unpaired *male → female* translation on the CelebA 128 × 128 dataset.

| Metric | Low stochasticity | | | High stochasticity | | |
|---|---|---|---|---|---|---|
| | CSBM $\alpha = 0.005$ | ASBM $\epsilon = 1$ | DSBM $\epsilon = 1$ | CSBM $\alpha = 0.01$ | ASBM $\epsilon = 10$ | DSBM $\epsilon = 10$ |
| FID ($\downarrow$) | **10.60** | 16.86 | 24.06 | **14.68** | 17.44 | 92.15 |
| CMMD ($\downarrow$) | **0.165** | 0.216 | 0.365 | **0.212** | 0.231 | 1.140 |
| LPIPS ($\downarrow$) | **0.175** | 0.242 | 0.246 | **0.170** | 0.294 | 0.386 |

The standard FID (Heusel et al., 2017), CMMD (Jayasumana et al., 2024), and LPIPS (Zhang et al., 2018) metrics comparison in Table 2 quantitatively demonstrates that our approach achieves better results than the other methods on the test set. Still, it is important to note that our experiments are conducted with $N = 100$ in D-IMF, which is higher than the $N = 3$ used in continuous-space D-IMF in ASBM, i.e., the trade-off between the number of time steps $N$ and the generation quality should be taken into account.

## Acknowledgements

The work was supported by the grant for research centers in the field of AI provided by the Ministry of Economic Development of the Russian Federation in accordance with the agreement 000000C313925P4F0002 and the agreement with Skoltech №139-10-2025-033.

## Impact Statement

This paper presents work whose goal is to advance the field of Machine Learning. There are many potential societal consequences of our work, none of which we feel must be specifically highlighted here.

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

## A    Limitations

One limitation of the proposed algorithm stems from the factorization of the transitional probabilities (see §3.2.3). This simplification comes at the cost of losing some information, as dependencies between features at the same step are not explicitly accounted for. However, it should be taken into account that this limitation is inherent to the most modern flow-based (Campbell et al., 2024; Gat et al., 2024) and diffusion-based (Hoogeboom et al., 2021; Austin et al., 2021) methods for discrete data. Recent approaches aim to address this issue by modeling the transition joint distribution using copulas (Liu et al., 2024) or energy functions (Xu et al., 2024). Additionally, in the Colored MNIST experiments (§4.3) the reference process varies slightly with $N$, due to implementation specifics. The MSE between cumulative transition matrices is bounded by $10^{-4}$, confirming that the induced discrepancies are statistically insignificant. Thus, these differences are negligible and do not impact the experiment's goals or broader implications.

## B    Proofs

*Proof of Theorem 3.1.* As stated in the theorem, we consider a process $q(x_0, x_{\mathrm{in}}, x_1) \in \Pi_N(p_0, p_1)$ with $N \geq 1$ intermediate time steps that is both Markov and reciprocal and a reference Markov process $q^{\mathrm{ref}} \in \mathcal{M}(\mathcal{X}^{N+2})$. We focus on the joint distribution of the boundary elements $x_0, x_1$, and a selected intermediate state $x_{t_n}$, where $n \in [1, N]$. This distribution, $p(x_0, x_{t_n}, x_1)$, can be expressed in two equivalent ways using the Markov or the reciprocal properties:

$$\underbrace{q(x_0, x_1) q^{\mathrm{ref}}(x_{t_n}|x_0, x_1)}_{\text{by reciprocal property}} = q(x_0, x_{t_n}, x_1) = \underbrace{p(x_0) q(x_{t_n}|x_0) q(x_1|x_{t_n})}_{\text{by Markov property}}.$$

Rearranging this equation and applying the logarithm thus we get:

$$\log q(x_1|x_0) = \log q(x_t|x_0) + \log q(x_1|x_{t_n}) - \log q^{\mathrm{ref}}(x_{t_n}|x_0, x_1).$$

Note that all the probability terms are strictly positive by the theorem's assumption. The knowledge that the last term $\log q^{\mathrm{ref}}(x_{t_n}|x_0, x_1)$ is Markov leads to following equation:

$$\log q(x_1|x_0) = \log q(x_{t_n}|x_0) + \log q(x_1|x_{t_n}) - \log \left( \frac{q^{\mathrm{ref}}(x_0) q^{\mathrm{ref}}(x_{t_n}|x_0) q^{\mathrm{ref}}(x_1|x_{t_n})}{q^{\mathrm{ref}}(x_0, x_1)} \right) =$$

$$= \underbrace{\log q(x_{t_n}|x_0) - \log q^{\mathrm{ref}}(x_{t_n}|x_0) - \log q^{\mathrm{ref}}(x_0)}_{\stackrel{\text{def}}{=} f_0(x_0, x_{t_n})} + \underbrace{\log q(x_1|x_{t_n}) - \log q^{\mathrm{ref}}(x_1|x_{t_n})}_{\stackrel{\text{def}}{=} f_1(x_{t_n}, x_1)} + \log q^{\mathrm{ref}}(x_0, x_1).$$

Thus, we get:

$$f(x_0, x_1) \stackrel{\text{def}}{=} \log q(x_1|x_0) - \log q^{\mathrm{ref}}(x_0, x_1) = f_0(x_0, x_{t_n}) + f_1(x_{t_n}, x_1). \tag{11}$$

Notably, $f(x_0, x_1)$ can be represented as a sum of two single-variable functions, $g_0(x_0)$ and $g_1(x_1)$. This could be observed by setting $x_1 = x^\dagger$ in (11), where $x^\dagger \in \mathcal{X}$ is some fixed point in the state space. Indeed, we have:

$$f(x_0, x_1) - f(x_0, x^\dagger) = \cancel{f_0(x_0, x_{t_n})} + f_1(x_{t_n}, x_1) - \cancel{f_0(x_0, x_{t_n})} - f_1(x_{t_n}, x^\dagger) = f_1(x_{t_n}, x_1) - f_1(x_{t_n}, x^\dagger).$$

Fixing $x_1 = x^\dagger$ makes $f(x_0, x^\dagger)$ depend only on $x_0$, so, we define $g_0(x_0) \stackrel{\text{def}}{=} f(x_0, x^\dagger)$. Likewise, with fixed $x_{t_n}$, the difference $f(x_0, x_1) - f(x_0, x^\dagger)$ depends only on $x_1$. Thus, we set $g_1(x_1) \stackrel{\text{def}}{=} f(x_0, x_1) - f(x_0, x^\dagger)$. Finally, we obtain:

$$\log q(x_1|x_0) = g_0(x_0) + g_1(x_1) + \log q^{\mathrm{ref}}(x_0, x_1).$$

Exponentiating both sides and multiplying by $p(x_0)$, we derive:

$$q(x_0, x_1) = \underbrace{e^{g_0(x_0)}}_{\psi(x_0)} q^{\mathrm{ref}}(x_1|x_0) \underbrace{e^{g_1(x_1)}}_{\phi(x_1)}.$$

According to (Léonard, 2013, Theorem 2.8), this formulation describes the optimal transport plan $q^*$ for the Static Schrödinger Bridge problem between $p_0$ and $p_1$. Alternatively, this can be derived as in (Gushchin et al., 2024b). Given that the assumption of the theorem ensures $q(x_{\mathrm{in}}|x_0, x_1) = q^{\mathrm{ref}}(x_{\mathrm{in}}|x_0, x_1)$, it follows that $q(x_0, x_{\mathrm{in}}, x_1)$ is a dynamic Schrödinger Bridge $q^*(x_0, x_{\mathrm{in}}, x_1)$. □

*Proof of Proposition 3.3.* Thanks to (Gushchin et al., 2024b, Proposition 3.5), it is known that

$$[proj_{\mathcal{M}}(q)](x_0, x_{\text{in}}, x_1) = \underset{m \in \mathcal{M}(\mathcal{X}^{N+2})}{\operatorname{argmin}} \text{KL}\left(q(x_0, x_{\text{in}}, x_1) \| m(x_0, x_{\text{in}}, x_1)\right), \tag{12}$$

where $q \in \mathcal{R}^{\text{ref}}(\mathcal{X}^{N+2})$ is a reciprocal process. Thus, we can decompose this KL divergence as follows:

$$\text{KL}\left(q(x_0, x_{\text{in}}, x_1) \| m(x_0, x_{\text{in}}, x_1)\right) = \mathbb{E}_{q(x_0, x_{\text{in}}, x_1)} \log \frac{q(x_0, x_{\text{in}}, x_1)}{m(x_0, x_{\text{in}}, x_1)}$$

$$= \mathbb{E}_{q(x_0, x_{\text{in}}, x_1)} \log \frac{p_0(x_0) q(x_1|x_0) q^{\text{ref}}(x_{\text{in}}|x_0, x_1)}{m(x_0) m(x_1|x_{t_N}) \prod_{n=1}^{N} m(x_{t_n}|x_{t_{n-1}})}. \tag{13}$$

Here, the denominator can be represented this way because $m$ is a Markov process, while the numerator is expressed using the reciprocal property of $q$. Next, we separate the corresponding colored terms, leading to:

$$(13) = \underbrace{-\mathbb{E}_{q(x_0, x_{t_N}, x_1)}\left[\log m(x_1|x_{t_N})\right]}_{L_1} + \mathbb{E}_{q(x_0, x_{\text{in}}, x_1)} \log \frac{\prod_{n=1}^{N} q^{\text{ref}}(x_{t_n}|x_{t_{n-1}}, x_1)}{\prod_{n=1}^{N} m(x_{t_n}|x_{t_{n-1}})} +$$

$$+ \underbrace{\text{KL}\left(p_0(x_0)\|m(x_0)\right)}_{L_0} + \underbrace{\mathbb{E}_{q(x_1, x_0)}\left[\log q(x_1|x_0)\right]}_{C_1}. \tag{14}$$

Rewriting the product inside the logarithm (violet term) as a sum of KL divergences, we obtain the following equation:

$$(14) = L_1 + \sum_{n=1}^{N} \mathbb{E}_{q(x_1, x_{t_{n-1}})} \text{KL}\left(q^{\text{ref}}(x_{t_n}|x_{t_{n-1}}, x_1)\|m(x_{t_n}|x_{t_{n-1}})\right) + L_0 + C_1. \tag{15}$$

We observe that, by construction, the Markov process $m$ preserves the terminal distribution when represented in a forward manner (5), i.e., $m(x_0) = p_0(x_0)$. Consequently, $L_0$ can be omitted since $\text{KL} = 0$, which completes the proof:

$$(15) = L_1 + \sum_{n=1}^{N} \mathbb{E}_{q(x_1, x_{t_{n-1}})} \text{KL}(q^{\text{ref}}(x_{t_n}|x_{t_{n-1}}, x_1)\|m(x_{t_n}|x_{t_{n-1}})) + C_1. \tag{16}$$

Additionally, because the Markovian projection (5) leaves the neighbouring-time joint distribution $q(x_{t_{n-1}}, x_{t_n})$ unchanged, we can train $m$ with the alternative objective:

$$\text{KL}\left(q(x_0, x_{\text{in}}, x_1)\|m(x_0, x_{\text{in}}, x_1)\right) = \mathbb{E}_{q(x_0, x_{\text{in}}, x_1)} \log \frac{q(x_0, x_{\text{in}}, x_1)}{m(x_0, x_{\text{in}}, x_1)} =$$

$$= \sum_{n=1}^{N+1} \mathbb{E}_{q(x_{t_{n-1}})} \text{KL}\left(q(x_{t_n}|x_{t_{n-1}})\|m(x_{t_n}|x_{t_{n-1}})\right) + \underbrace{\text{KL}\left(p_0(x_0)\|m(x_0)\right)}_{L_0}. \tag{17}$$

Similarly, we discard $L_0$, leaving us with an objective that minimizes the divergence between one-step transition probabilities of the given process $q$ and the desired Markov process $m$. $\qquad\square$

# C  Additional Experiments

## C.1  Alternative Losses

Proposition 3.3 shows that two equivalent KL-based training objectives yield the same optimal solution. This naturally suggests a generalization to a broader class of divergences $D$.

**The Original Objective.**    First, let us consider the original objective function given in (16). To ensure that substituting an alternative divergence does not alter its minima, the replacement must be equivalent in this context. Specifically, the $L_1$ term can be reformulated as the KL divergence between a Kronecker delta distribution and the transition distribution of $m$, i.e.:

$$L_1 = -\mathbb{E}_{q(x_0,x_{t_N},x_1)}\left[\log m(x_1|x_{t_N})\right] = \mathbb{E}_{q(x_0,x_{t_N},x_1)}\mathbb{E}_{\delta_{x_1}(\widetilde{x}_1)}\left[\log \frac{\delta_{x_1}(\widetilde{x}_1)}{m(\widetilde{x}_1|x_{t_N})}\right] =$$

$$= \mathbb{E}_{q(x_0,x_{t_N},x_1)}\mathrm{KL}\left(\delta_{x_1}(\widetilde{x}_1)\|m(\widetilde{x}_1|x_{t_N})\right) = \mathbb{E}_{q(x_0,x_{t_N},x_1)}\mathrm{KL}\left(q(x_1|x_{t_N},x_1)\|m(\widetilde{x}_1|x_{t_N})\right).$$

Consequently, the $L_1$ term can be moved under the sum of the violet term, leading to:

$$(16) = \sum_{n=1}^{N+1}\mathbb{E}_{q(x_1,x_{t_{n-1}})}\mathrm{KL}\left(q^{\mathrm{ref}}(x_{t_n}|x_{t_{n-1}},x_1)\|m(x_{t_n}|x_{t_{n-1}})\right) + C_1.$$

By restricting the choice of divergences to the Bregman family, we ensure that the minimum is attained at the same value, namely, $\mathbb{E}_{q(x_1|x_{t_{n-1}})}\left[q^{\mathrm{ref}}(x_{t_n}|x_{t_{n-1}},x_1)\right] = q(x_{t_n}|x_{t_{n-1}})$ (Banerjee et al., 2005). Thus, any Bregman divergence can be used as the objective. As an example, we consider the MSE loss as an alternative to the KL divergence:

$$\operatorname*{argmin}_{m\in\mathcal{M}(\mathcal{X}^{N+2})} \sum_{n=1}^{N+1}\mathbb{E}_{q(x_1,x_{t_{n-1}})}\mathrm{KL}\left(q^{\mathrm{ref}}(x_{t_n}|x_{t_{n-1}},x_1)\|m(x_{t_n}|x_{t_{n-1}})\right) =$$

$$= \operatorname*{argmin}_{m\in\mathcal{M}(\mathcal{X}^{N+2})} \sum_{n=1}^{N+1}\mathbb{E}_{q(x_1,x_{t_{n-1}})}\left[q^{\mathrm{ref}}(x_{t_n}|x_{t_{n-1}},x_1) - m(x_{t_n}|x_{t_{n-1}})\right]^2 \quad (18)$$

Applying this loss parametrization from §3.2.3 and repeating the derivation leads to the following objectives:

$$L_{\mathrm{MSE}}(\theta) = \sum_{n=1}^{N+1}\mathbb{E}_{q(x_0,x_1)}\mathbb{E}_{q^{\mathrm{ref}}(x_{t_{n-1}}|x_0,x_1)}\left[q^{\mathrm{ref}}(x_{t_n}|x_{t_{n-1}},x_1) - \mathbb{E}_{\widetilde{q}_\theta(\widetilde{x}_1|x_{t_{n-1}})}[q^{\mathrm{ref}}(x_{t_n}|x_{t_{n-1}},\widetilde{x}_1)]\right]^2, \quad (19)$$

$$L_{\mathrm{MSE}}(\eta) = \sum_{n=1}^{N+1}\mathbb{E}_{q(x_0,x_1)}\mathbb{E}_{q^{\mathrm{ref}}(x_{t_n}|x_0,x_1)}\left[q^{\mathrm{ref}}(x_{t_{n-1}}|x_{t_n},x_0) - \mathbb{E}_{\widetilde{q}_\eta(\widetilde{x}_0|x_{t_n})}[q^{\mathrm{ref}}(x_{t_{n-1}}|x_{t_n},\widetilde{x}_0)]\right]^2, \quad (20)$$

for forward and backward parametrization, respectively. To test the MSE loss, we repeat the 2D domain translation experiment between the *Gaussian* and *Swiss-Roll* distributions. It could be observed that the generated samples and trajectories with the MSE loss in Figure 5 appear visually similar to those obtained using the KL loss shown in Figure 2.

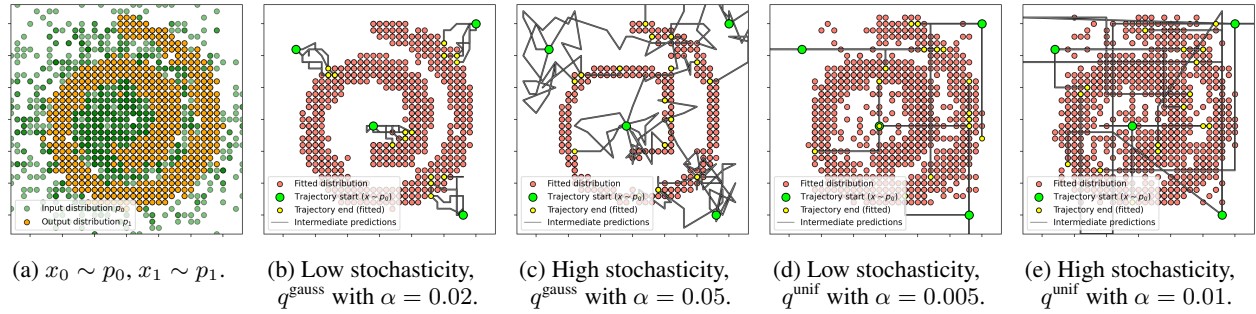

(a) $x_0 \sim p_0$, $x_1 \sim p_1$.  (b) Low stochasticity, $q^{\mathrm{gauss}}$ with $\alpha = 0.02$.  (c) High stochasticity, $q^{\mathrm{gauss}}$ with $\alpha = 0.05$.  (d) Low stochasticity, $q^{\mathrm{unif}}$ with $\alpha = 0.005$.  (e) High stochasticity, $q^{\mathrm{unif}}$ with $\alpha = 0.01$.

*Figure 5.* SB between 2D *Gaussian* and *Swiss-Roll* distributions learned by our CSBM algorithm with MSE loss in Equations (19) and (20) for different reference processes $q^{\mathrm{unif}}$ and $q^{\mathrm{gauss}}$ with varying parameters $\alpha$.

**The Alternative Objective.**    Analogous reasoning extends to the alternative objective in (17). Although the conditional distribution $q(x_{t_n}|x_{t_{n-1}})$ is generally unavailable in closed form, it can be sampled. This property suggests employing an adversarial training strategy, following the approach in (Gushchin et al., 2024b).

## C.2 Unpaired Translation on Colored MNIST with $q^{\text{unif}}$

We perform an additional Colored-MNIST experiment using a uniform reference process $q^{\text{unif}}$. Here we set $N = 25$ and test $\alpha \in \{0.01, 0.05\}$. Mini-batch OT is not applied at the D-IMF 1 iteration. The samples in Figure 6 demonstrate the failure to match the digit colors, showing that a uniform transition matrix is not suitable for this domain.

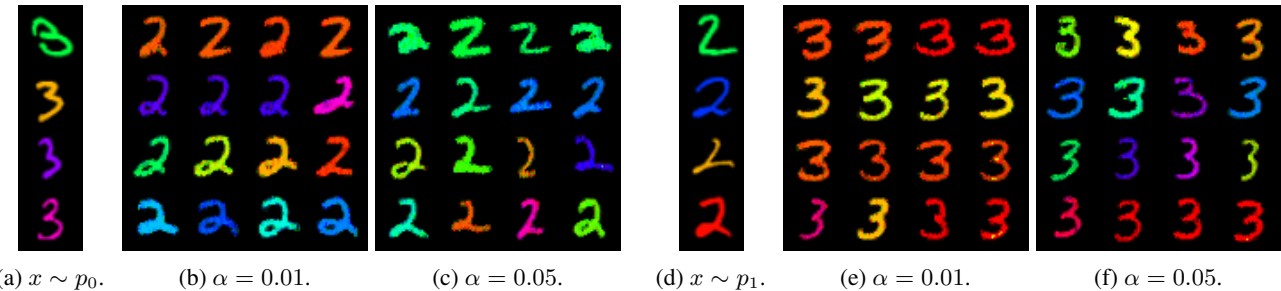

(a) $x \sim p_0$.  (b) $\alpha = 0.01$.  (c) $\alpha = 0.05$.  (d) $x \sim p_1$.  (e) $\alpha = 0.01$.  (f) $\alpha = 0.05$.

*Figure 6.* Results of colored digits unpaired translation learned by our CSBM algorithm with reference process $q^{\text{unif}}$ and varying stochasticity parameter $\alpha$.

## C.3 Continuous Methods in Latent Space

For completeness, we also trained DSBM in the latent space. For a fair comparison, we train DSBM on the same latent space used for CSBM, following the approach in (Rombach et al., 2022, Appendix G). Concretely, because the decoder expects discrete tokens, our pipeline proceeds as follows: (1) map the images to their continuous latent representations, (2) apply DSBM in this continuous space, (3) vector-quantize the resulting latents, and (4) pass the quantized tokens through the decoder. Unfortunately, the results are not satisfactory, as the model tended to collapse to the identity mapping with $\epsilon = 1$ and $\epsilon = 10$ (see Figure 7). Due to these limitations, we do not proceed with training ASBM and choose not to compare both methods with CSBM in such settings. One may ask why CSBM performs better in this setting. We hypothesize that this is due to the choice of the reference process $q^{\text{unif}}$, which is better suited to the VQ-GAN latent space.

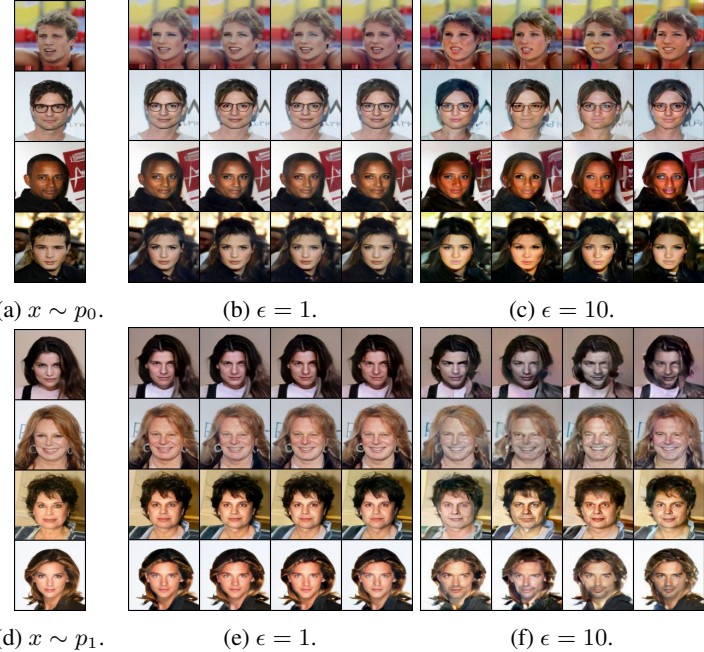

(a) $x \sim p_0$.  (b) $\epsilon = 1$.  (c) $\epsilon = 10$.

(d) $x \sim p_1$.  (e) $\epsilon = 1$.  (f) $\epsilon = 10$.

*Figure 7.* Results of training DSBM (Shi et al., 2023) on VQ-GAN lantent space of CelebA. The VQ-GAN model is the same as in the main experiments (§4.4).

*Table 3.* Metrics comparison of CSBM (**ours**), CAAE (Shen et al., 2017), Del.&Ret. (Li et al., 2018), Seq2SentiSeq (Luo et al., 2019), BST (Prabhumoye et al., 2018), FGIM (Wang et al., 2019), PST (He et al., 2020) and SCT$_1$ (Mukherjee et al., 2022) for unpaired *negative* $\leftrightarrow$ *positive* style transfer on the Amazon Reviews dataset. Bold denotes the best value, and underline the second best. Metrics of baseline methods are taken from (Mukherjee et al., 2022) and marked with a superscript †.

| Metric | CSBM $\alpha = 0.005$ | CSBM $\alpha = 0.01$ | CAAE† | Del.&Ret.† | Seq2SentiSeq† | BST† | FGIM† | PST† | SCT$_1$† |
|---|---|---|---|---|---|---|---|---|---|
| Accuracy (↑) | 79.3 | 76.5 | 88.6 | 69.9 | 92.4 | **93.5** | 79.3 | 91.5 | 82.0 |
| NLL (↓) | **5.4** | **5.4** | 74.0 | 85.1 | 42.0 | 61.0 | 116.8 | 65.9 | 79.6 |
| BLEU (↑) | 72.5 | **74.8** | 3.2 | 14.7 | 0.0 | 0.9 | 10.6 | 9.5 | 13.7 |

*Table 4.* Style transfers of CSBM (**ours**), Del.&Ret. (Li et al., 2018), BST (Prabhumoye et al., 2018), FGIM (Wang et al., 2019), PST (He et al., 2020) and SCT$_1$ (Mukherjee et al., 2022) on Amazon Reviews dataset. Samples of baseline methods are taken from (Mukherjee et al., 2022) and marked with a superscript †.

| | *negative → positive* | *positive → negative* |
|---|---|---|
| **Source** | ***movie was a waste of money : this movie totally sucks .*** | ***my daughter loves them : )*** |
| CSBM $\alpha = 0.005$ | *movie was great value for the money : this movie totally wass .* | *my daughter hates them :(* |
| CSBM $\alpha = 0.01$ | *movie was great value for the money : this movie totally superb .* | *my daughter hates them :(* |
| Del.&Ret.† | *our favorite thing was a movie story : the dream class roll !* | *my daughter said i was still not acknowledged .* |
| BST† | *stan is always a great place to get the food .* | *do n't be going here .* |
| FGIM† | *movie is a delicious atmosphere of : this movie totally sucks movie !* | *i should not send dress after me more than she would said not ?* |
| PST† | *this theater was a great place , we movie totally amazing .* | *yup daughter has left ourselves .* |
| SCT$_1$† | *movie : a great deal of money : this movie is absolutely perfect .* | *my daughter hates it : my daughter .* |
| **Source** | ***nothing truly interesting happens in this book .*** | ***best fit for my baby : this product is wonderful ! !*** |
| CSBM $\alpha = 0.005$ | *everything truly interesting happens in this book .* | *not fit for my baby : this product is junk !!* |
| CSBM $\alpha = 0.01$ | *everything truly interesting happens in this book .* | *not fit for my baby : this product is bad !!* |
| Del.&Ret.† | *nothing truly interesting happens in this book .* | *my mom was annoyed with my health service is no notice .* |
| BST† | *very good for the best .* | *bad customer service to say the food , and it is n't .* |
| FGIM† | *nothing truly interesting happens in this book make it casual and spot .* | *do not buy my phone : this bad crap was worst than it ?* |
| PST† | *haha truly interesting happens in this book .* | *uninspired .* |
| SCT$_1$† | *in this book is truly a really great book .* | *not good for my baby : this product is great ! ! ! ! ! ! ! !* |

## C.4 Unpaired Text Style Transfer of Amazon Reviews

This section examines the text domain, focusing on style transfer in the Amazon Reviews corpus (Ni et al., 2019). The task is to convert reviews with *negative* sentiment into ones with *positive* sentiment and vice versa. We adopt the filtered, pre-processed split of (Mukherjee et al., 2022). Reviews are tokenized with a unigram SentencePiece model (Kudo & Richardson, 2018) that has a vocabulary size set to $S = 8\,192$. Each review is then padded or truncated to a fixed length of $D = 100$. We evaluate the uniform reference process $q^{\text{ref}}$ for $\alpha \in \{0.005, 0.01\}$. The reported scores are averaged over both transfer directions *negative* $\leftrightarrow$ *positive* and compared with baselines, using the metrics from (Mukherjee et al., 2022).

To mirror the image-domain protocol, we select analogous text metrics. Target alignment is measured with the Hugging Face pipeline's default sentiment classifier, complemented by the negative log-likelihood (NLL) under GPT-2 Large (Radford et al., 2019). Similarity between the transferred text and its source is measured with BLEU (Papineni et al., 2002). Quantitative metrics appear in Table 3, while representative samples are shown in Table 4.

CSBM excels at content preservation, achieving the highest BLEU score and the lowest NLL, indicating fluent, meaning-faithful rewrites. Its sentiment-transfer accuracy is lower than half of the methods, yet manual inspection of the samples in Table 4 suggests that most generations convey the correct polarity.

## D Practical Details

### D.1 Construction and Selection of Reference Processes $q^{\text{ref}}$

**Construction of $q^{\text{ref}}$.** The article touches only briefly on how the reference processes $q^{\text{ref}}$ are built. In the current scheme, $q^{\text{ref}}$ is assembled by chaining intermediate transition probabilities $q^{\text{ref}}(x_{t_n}|x_{t_{n-1}})$. Consequently, the full end-to-end transition $q^{\text{ref}}(x_1|x_0)$ varies with the choice of $\alpha$, the transition matrix $Q_n$, and the discretizations level $N$, rather than remaining fixed across settings. Due to this, for example, the increasing number of steps $N$ forces us to choose a smaller $\alpha$. If $\alpha$ remains too large, the overall transition probability $q^{\text{ref}}(x_1|x_0)$ converges to the stationary distribution, making every start state equally likely to reach every end state. A uniform distribution is not inherently wrong, but it defeats our aim, as we want $\alpha$ to control the overall stochasticity in the process. Thus, building a non-uniform, non-Gaussian $q^{\text{ref}}$ is considerably more challenging, prompting us to explore new construction strategies in the future.

**Selection of $\alpha$.** Across many experiments, we observed a pattern for choosing $\alpha$. Overall, the general idea follows the same intuition as choosing $\epsilon$ in continuous SB methods (Shi et al., 2023; Gushchin et al., 2024b). Specifically, lower values of $\alpha$ lead to less stochasticity in the trajectories, resulting in higher similarity to the input data but a lower-quality approximation of the target distribution. At very low values, the model may collapse due to insufficient stochasticity. Conversely, higher values of $\alpha$ introduce more variability, reducing similarity to the initial data. Beyond a certain point, excessively large values $\alpha$ make the model difficult to train, leading to a drop in both quality and similarity. Unfortunately, the effective range of these behaviors is highly dependent on the dataset and the chosen reference process. Nonetheless, we provide reasonable baseline values from which one can begin and adjust as needed.

### D.2 Loss Function of CSBM

In this section, we outline the optimization procedure for the parameterization in (9), obtained by substituting $m = q_\theta$ into (10). Following (Austin et al., 2021), we parameterize the model to predict the terminal point $x_1$ or $x_0$ for the forward or backward reparameterization, respectively, and adopt a hybrid loss that sums the base loss with the loss $L_{\text{simple}}$, scaled by a weighting factor $\lambda$. The resulting training objective is therefore given by:

$$
L(\theta) = \mathbb{E}_{q(x_0,x_1)}\Bigg[\sum_{n=1}^{N} \mathbb{E}_{q^{\text{ref}}(x_{t_{n-1}}|x_0,x_1)}
$$

$$
\text{KL}\left(q^{\text{ref}}(x_{t_n}|x_{t_{n-1}},x_1)\|\mathbb{E}_{\widetilde{q}_\theta(\widetilde{x}_1|x_{t_{n-1}})}[q^{\text{ref}}(x_{t_n}|x_{t_{n-1}},\widetilde{x}_1)]\right) - \lambda \overbrace{\log \widetilde{q}_\theta(\widetilde{x}_1|x_{t_{n-1}})}^{L_{\text{simple}}}
$$

$$
- \mathbb{E}_{q^{\text{ref}}(x_{t_N}|x_0,x_1)}\left[\log \widetilde{q}_\theta(x_1|x_{t_N})\right]\Bigg]. \quad (21)
$$

Since the backward decomposition of $m$ also holds for Proposition 10, we can apply a similar parametrization. In this case, we use a neural network with parameters $\eta$ to predict $x_0$:

$$
L(\eta) = \mathbb{E}_{q(x_0,x_1)}\Bigg[\sum_{n=2}^{N+1} \mathbb{E}_{q^{\text{ref}}(x_{t_n}|x_0,x_1)}
$$

$$
\text{KL}\left(q^{\text{ref}}(x_{t_{n-1}}|x_{t_n},x_0)\|\mathbb{E}_{\widetilde{q}_\eta(\widetilde{x}_0|x_{t_n})}[q^{\text{ref}}(x_{t_{n-1}}|x_{t_n},\widetilde{x}_0)]\right) - \lambda \overbrace{\log \widetilde{q}_\eta(\widetilde{x}_0|x_{t_n})}^{L_{\text{simple}}}
$$

$$
- \mathbb{E}_{q^{\text{ref}}(x_{t_1}|x_0,x_1)}\left[\log \widetilde{q}_\eta(x_0|x_{t_1})\right]\Bigg]. \quad (22)
$$

For further details on the training process, we refer the reader to (Austin et al., 2021).

### D.3 Training Aspects

For the implementation of the training logic, we use the official D3PM repository (Austin et al., 2021) as a reference:

https://github.com/google-research/google-research/tree/master/d3pm

*Table 5.* Hyperparameters for experiments. Lr denotes the learning rate, and *m* represents millions. Params indicate the number of model parameters, where for the CelebA dataset, the first value corresponds to the model and the second to the VQ-GAN.

| Experiment | Initial coupling | D-IMF outer iterations | D-IMF=1 grad updates | D-IMF grad updates | $N$ | Batch size | Lr | Params |
|---|---|---|---|---|---|---|---|---|
| 2D | Ind | 10 | 400 000 | 40 000 | 10 | 512 | 0.0004 | 46588 |
| Colored MNIST | MB | 3 | 200 000 | 40 000 | 2, 4, 10, 25, 50, 100 | 128 | 0.0002 | 34m |
| CelebA | Ind | 4 | 800 000 | 40 000 | 100 | 32 | 0.0004 | 93m + 70m |
| Amazon Reviews | Ind | 5 | 800 000 | 40 000 | 100 | 32 | 0.0004 | 100m |

**Shared Aspects.** For all experiments, we use the AdamW optimizer with fixed betas of 0.95 and 0.99. Additionally, we apply Exponential Moving Average (EMA) smoothing to stabilize training and enhance final model performance. The EMA decay rate is consistently tuned across all experiments and set to 0.999, except for the Colored MNIST experiment, where it is set to 0.9999. For all experiments, we set the weighting factor of $L_{\text{simple}}$ to 0.001.

For the 2D and colored MNIST experiment, we follow the preprocessing approach from (Austin et al., 2021), where the logits of $q_\theta(\widetilde{x}_1|x_{t_{n-1}})$ are modeled directly as the output of a neural network.

Notably, various previous works have introduced different initial couplings $q^0(x_0, x_1)$, such as the standard independent coupling $p_0(x_0)p_1(x_1)$ (Shi et al., 2023; Gushchin et al., 2024b), couplings derived from a reference process, e.g., $p_0(x_0)q^{\text{ref}}(x_1|x_0)$ (Shi et al., 2023), and mini-batch OT couplings referred as MB, i.e., discrete Optimal Transport solved on mini-batch samples (Tong et al., 2024). For a more comprehensive overview of coupling strategies, see (Kholkin et al., 2024). In this work, we focus exclusively on the independent and mini-batch initial coupling.

**Experiment-specific Aspects.** For the **2D experiment** (§4.2), we use a simple MLP model with hidden layers of size $[128, 128, 128]$ and ReLU activations. To condition on time, we use a simple lookup table, i.e., an embedding layer of size 2.

For the **colored MNIST experiment** (§4.3), we follow (Austin et al., 2021) and use an architecture based on a PixelCNN++ backbone (Salimans et al., 2016), utilizing a U-Net (Ronneberger et al., 2015) with a ResNet-like structure. The model operates at four feature map resolutions, with two convolutional residual blocks per resolution level and a channel multiplier of $(1, 2, 2, 2)$. At the $16 \times 16$ resolution level, a self-attention block is incorporated between the convolutional blocks. For time encoding, we apply Transformer sinusoidal position embeddings to each residual block. We train the model on a training subset of size 60 000 and generate images from the hold-out set.

For the **CelebA experiment** (§4.4), we employ VQ-Diffusion (Gu et al., 2022), which consists of two models: VQ-GAN (Esser et al., 2021) and a transformer-based diffusion model. The VQ-GAN component is trained using the official repository:



https://github.com/CompVis/taming-transformers.



We slightly modify the experimental setup of unconditional generation for CelebA-HQ from (Esser et al., 2021) by reducing the number of resolution levels to three, with scaling factors of $(1, 2, 4)$. This adjustment accounts for our use of CelebA at $128 \times 128$ resolution, compared to $256 \times 256$ in CelebA-HQ. The discrete diffusion model is adopted from:



https://github.com/microsoft/VQ-Diffusion.



Our diffusion model consists of multiple transformer blocks, each incorporating full attention and a feed-forward network (FFN). We follow the small model configuration from (Gu et al., 2022), which consists of 18 transformer blocks with an increased channel size of 256. The FFN is implemented using two convolutional layers with a kernel size of 3, and the channel expansion rate is set to 2. Additionally, we inject time step information through the AdaLN operator.

We train the model on 162 770 pre-quantized images of celebrities. For evaluation, we compute FID and CMMD using 11 816 hold-out images to ensure consistency with the evaluation protocol from (Gushchin et al., 2024b). Likewise, the images presented in the main text of the paper are generated using this hold-out set.

For the **Amazon experiment** (Appendix C.4), we train a unigram SentencePiece tokenizer (Kudo & Richardson, 2018) that includes explicit start-of-sentence (``) and padding (`<pad>`) tokens, following the procedure of (Austin et al., 2021). The backbone is the DiT model (Peebles & Xie, 2023), with the implementation available at:

https://github.com/kuleshov-group/mdlm.

We employ the "small" variant with 12 transformer blocks, each with a hidden size of 768 and 12 attention heads. Every block contains multi-head self-attention, rotary positional embeddings, and an MLP with a dropout rate of 0.1. Noise-level information is injected via a 128-dimensional AdaLN modulation vector. The model is trained on $104\,000$ pre-tokenized reviews and evaluated on $2\,000$ reviews from the held-out test set. The rest hyperparameters are presented in Table 5.

**Computational Time.** Training the 2D experiment requires several hours on a single A100 GPU. The colored MNIST experiment takes approximately two days to train using two A100 GPUs. The most computationally demanding task, the CelebA and Amazon Reviews experiments, requires around five days of training on four A100 GPUs.

## D.4 Additional Images

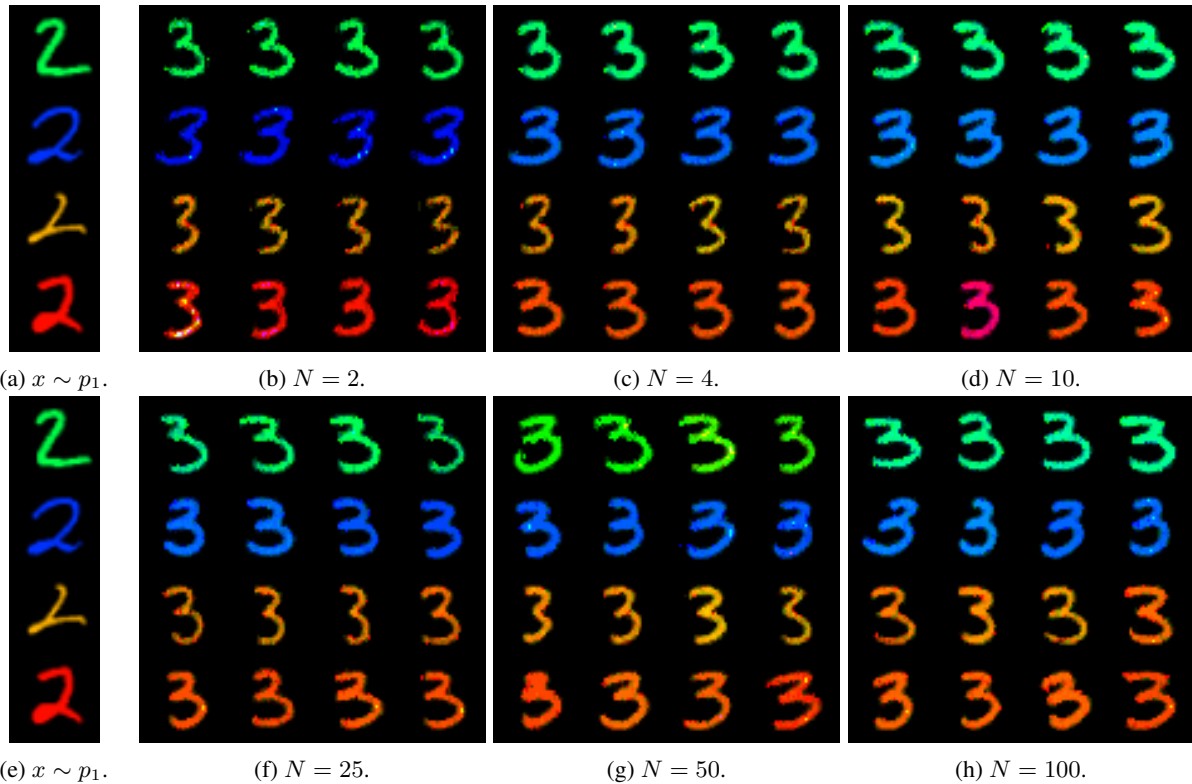

(a) $x \sim p_1$.  (b) $N = 2$.  (c) $N = 4$.  (d) $N = 10$.

(e) $x \sim p_1$.  (f) $N = 25$.  (g) $N = 50$.  (h) $N = 100$.

*Figure 8.* Results of colored digits unpaired translation "2" → "3" learned by our CSBM algorithm with reference process $q^{\text{gauss}}$ and varying number of time moments $N$.

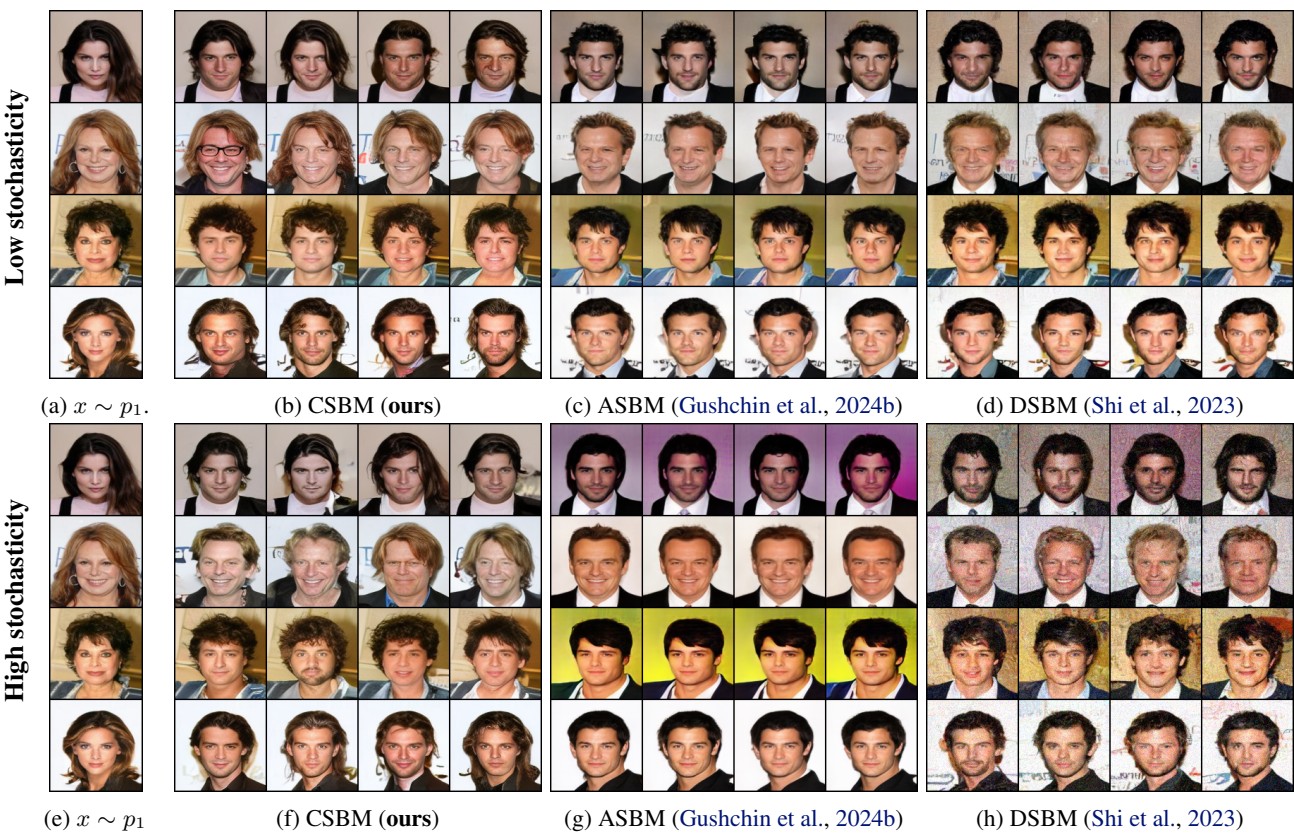

(a) $x \sim p_1$.    (b) CSBM (**ours**)    (c) ASBM (Gushchin et al., 2024b)    (d) DSBM (Shi et al., 2023)

(e) $x \sim p_1$    (f) CSBM (**ours**)    (g) ASBM (Gushchin et al., 2024b)    (h) DSBM (Shi et al., 2023)

*Figure 9.* Comparison of *female* → *male* translation on the CelebA $128 \times 128$ dataset using CSBM (ours), ASBM, and DSBM. The low-stochasticity setting for CSBM corresponds to $\alpha = 0.005$, while the high-stochasticity setting corresponds to $\alpha = 0.01$. The stochasticity parameters for ASBM and DSBM are taken from (Gushchin et al., 2024b).

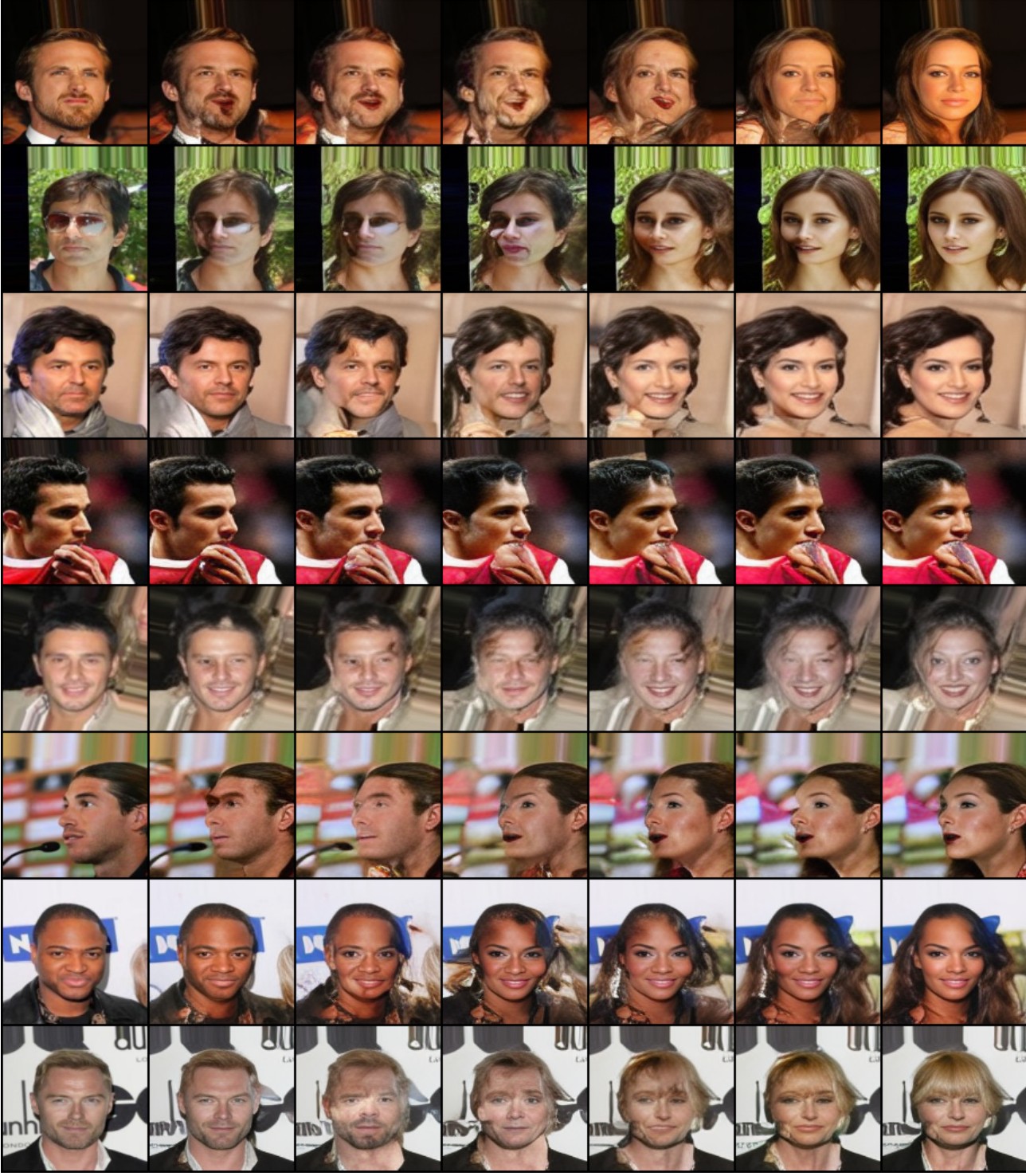

*Figure 10. male → female* translation trajectories on the CelebA $128 \times 128$ dataset using CSBM with $\alpha = 0.01$. Each column corresponds to time moments 0, 10, 25, 50, 75, 90, and 101.

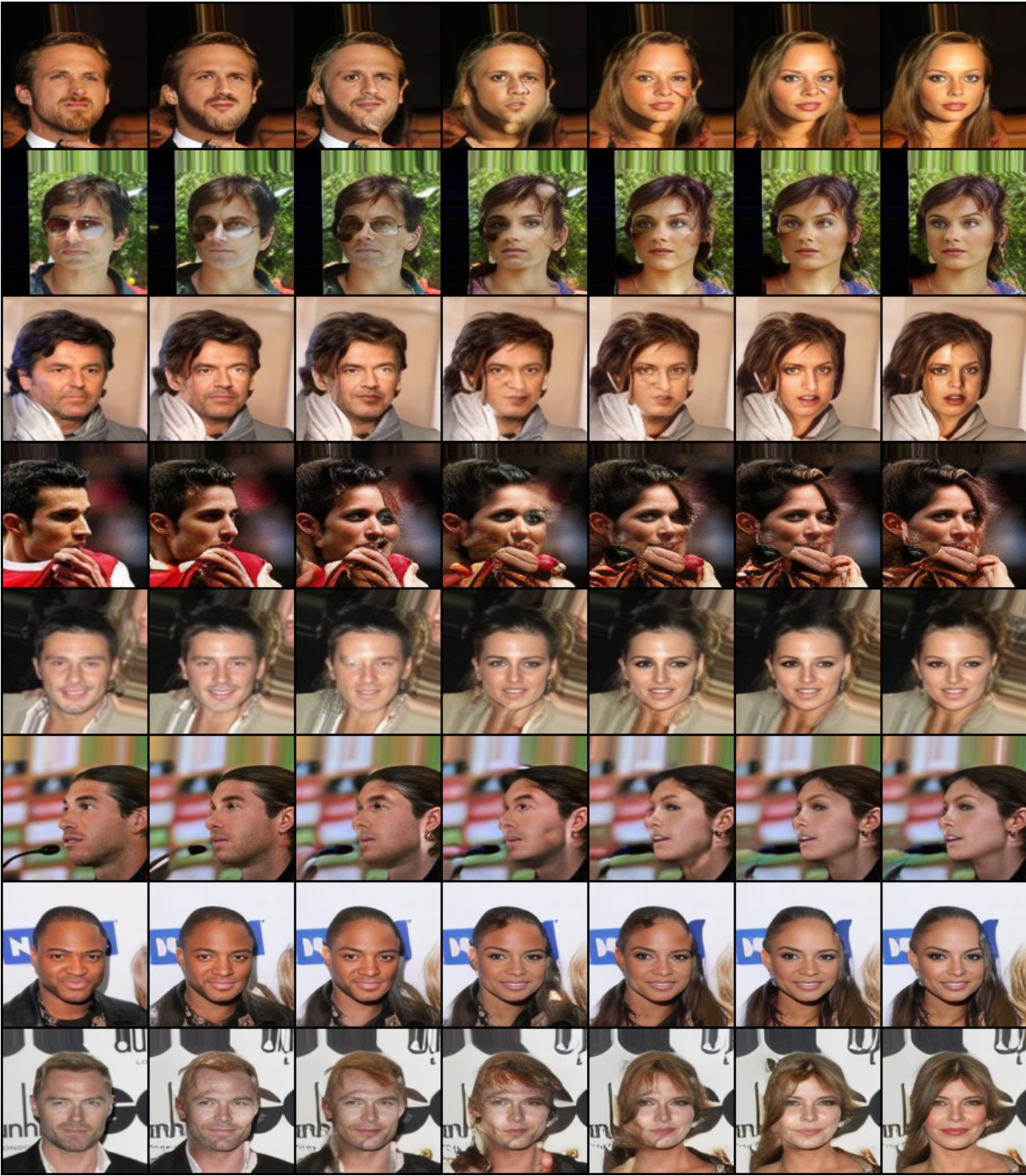

*Figure 11. male → female* translation trajectories on the CelebA $128 \times 128$ dataset using CSBM with $\alpha = 0.005$. Each column corresponds to time moments 0, 10, 25, 50, 75, 90, and 101.

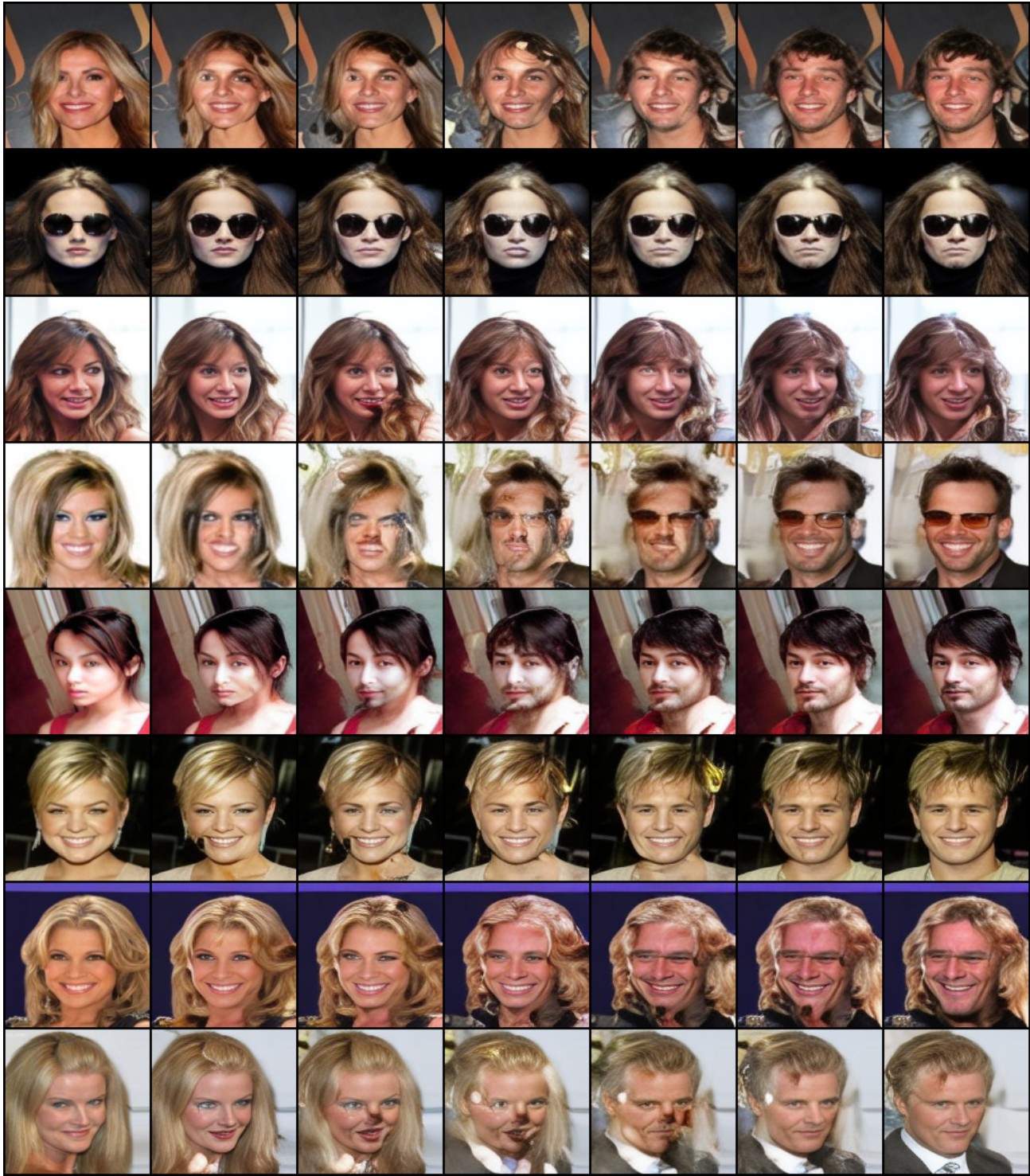

Figure 12. *female* → *male* translation trajectories on the CelebA $128 \times 128$ dataset using CSBM with $\alpha = 0.01$. Each column corresponds to time moments 0, 10, 25, 50, 75, 90, and 101.

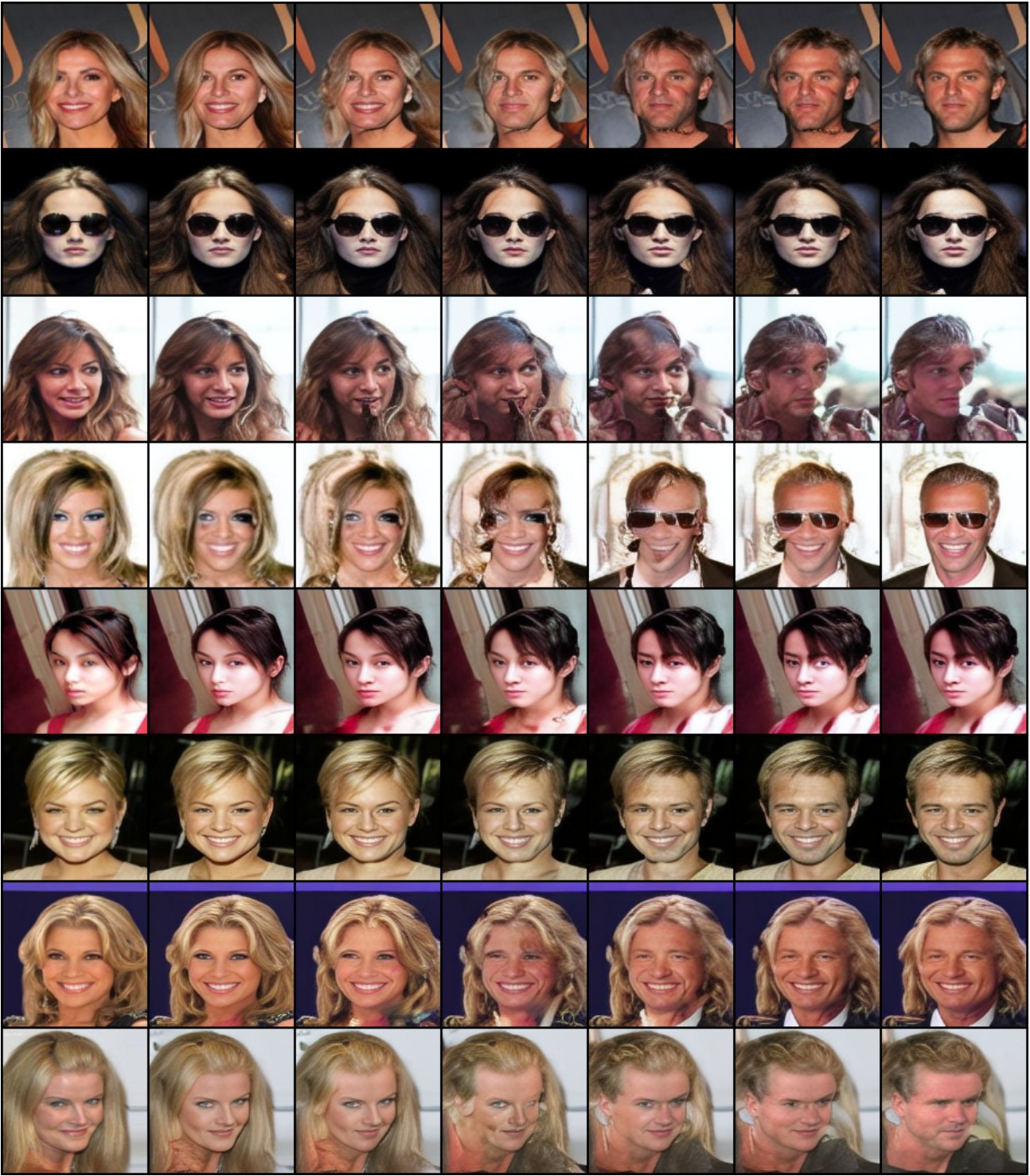

*Figure 13.* female → male translation trajectories on the CelebA 128 × 128 dataset using CSBM with α = 0.005. Each column corresponds to time moments 0, 10, 25, 50, 75, 90, and 101.

