# OpenReview forum: "Categorical Schrödinger Bridge Matching"
_ICML.cc/2025/Conference — ICML 2025 poster_

### Official Review · Reviewer_hwSE · 2025-03-07

**Overall Recommendation:** 4

**Summary:**

The authors:
- provide a proof for the convergence of discrete-time IMF in discrete-state spaces.
- develop an algorithm called "Categorical SBM" that approximates a solution to the SB problem for discrete-state spaces.

**Claims And Evidence:**

I'm not well equipped to answer this question.

**Essential References Not Discussed:**

I don't know.

**Experimental Designs Or Analyses:**

These experiments are standard.

**Methods And Evaluation Criteria:**

The experiments make sense to me (inter-domain translation of images), and the method performs on par with what was compared.

I don't know the literature enough to fully appreciate these results.

**Other Comments Or Suggestions:**

There are many typos regarding word order (see lines 112 and 116) as well as the use of the word "the".

They don't affect the understanding.

**Other Strengths And Weaknesses:**

The paper is very well written.

**Questions For Authors:**

None.

**Relation To Broader Scientific Literature:**

I don't know.

**Theoretical Claims:**

I checked the proofs to the best of my ability, and they look correct.

However, I can't provide any merit on how relevant it is to the current understanding of the topic.

---

> ### Author Rebuttal · Authors · 2025-04-01
>
> Thank you for your comments and positive evaluation. We will correct the typos you mentioned. If you're interested in the topic, we recommend checking the other reviews and these works on the Schrödinger Bridge Problem [1, 2, 3].
>
> [1] Kim, Jun Hyeong, et al. "Discrete Diffusion Schrödinger Bridge Matching for Graph Transformation." arXiv preprint arXiv:2410.01500 (2024).
>
> [2] Shi, Yuyang, et al. "Diffusion Schrödinger bridge matching." Advances in Neural Information Processing Systems 36 (2023): 62183-62223.
>
> [3] Gushchin, Nikita, et al. "Adversarial Schrödinger Bridge Matching." The Thirty-eighth Annual Conference on Neural Information Processing Systems.

---

### Official Review · Reviewer_1AGf · 2025-03-12

**Overall Recommendation:** 2

**Summary:**

The paper proposes an algorithm based on Iterative Markovian Fitting (IMF) for solving Schrödinger Bridge (SB) in discrete (categorical) space. The contribution of the paper therefore lies in the extension of SB, originally constructed in continuous state spaces, and its data-driven learning-based algorithm to discrete setup. Experiments are conducted on 2D synthetic dataset and latent-space images.

**Claims And Evidence:**

- Can the author clarify the difference between proposed method to DDSBM, which does present theoretical results for continuous-time IMF in discrete spaces? I'm fairly familiar with DDSBM and, since both are based on IMF, the methods seem to collapse in practice when time is discretized.

**Essential References Not Discussed:**

N

**Experimental Designs Or Analyses:**

- I'm not convinced by Table 2. The author should report ASBM and DSBM in a similar GAN-based continuous latent spaces for a fair comparison.

- It seems faulty to claim the dimensionality of CelebA Faces to be 1024^256. Given the factorized parametrization, the dimension that the proposed method handles should be 1024*256.

- Fig 3's caption should clarify that images are generated in VQ-GAN latent spaces. I think it's boarderline misleading to not mention specifically in the caption that it's a GAN-based latent-space image experiments. GAN's latent spaces are not only of lower dimension but also much structural.

**Methods And Evaluation Criteria:**

Y

**Other Comments Or Suggestions:**

N

**Other Strengths And Weaknesses:**

N

**Questions For Authors:**

My main questions and concerns, as listed above, are the similarity to DDSBM (ICLR'25) and experiment setups. The only comparison to prior SB works is Table 2 which is not conducted fairly.

**Relation To Broader Scientific Literature:**

Discrete space is an important extension of data-driven methods. Thm 3.1 could potentially handle other data types/spaces, which may also be of independent interests.

**Theoretical Claims:**

Y

---

> ### Author Rebuttal · Authors · 2025-04-01
>
> Dear reviewer 1AGf thank you for your questions and commentaries.
>
> **[Q. 1] Can the author clarify the difference between proposed method to DDSBM, which does present theoretical results for continuous-time IMF in discrete spaces? I'm fairly familiar with DDSBM and, since both are based on IMF, the methods seem to collapse in practice when time is discretized.**
>
> The answer to this question can be found in our response to reviewer [MBG2](https://openreview.net/forum?id=RBly0nOr2h&noteId=20eaTHc0ic) [W. 1] and [W. 2].
>
> **[W. 1] I'm not convinced by Table 2. The author should report ASBM and DSBM in a similar GAN-based continuous latent spaces for a fair comparison.**
>
> Regarding the setup on the CelebA dataset, we agree with your concerns. We did attempt to train DSBM in the latent space. For a fair comparison, we ran DSBM on the same latent space used for CSBM, following the approach in [1, Appendix G]. However, the results were not satisfactory, as the model tended to collapse to the identity mapping with $\epsilon = 1$ and $\epsilon =  3$ (**LINK:** see [figures](https://anonymous.4open.science/r/images-64B3/) with prefix 'latent'). Due to these limitations, we did not proceed with training ASBM and chose not to compare both methods with CSBM in such settings. One may ask why CSBM performs better in this setting. We hypothesize that this is due to the choice of the reference process, with $q^{\text{unif}}$ being more suitable for the latent space of VQ-GAN.
>
> [1] Rombach, Robin, et al. "High-resolution image synthesis with latent diffusion models." Proceedings of the IEEE/CVF conference on computer vision and pattern recognition. 2022.
>
> **[W. 2] It seems faulty to claim the dimensionality of CelebA Faces to be $1024^{256}$. Given the factorized parametrization, the dimension that the proposed method handles should be $1024*256$.**
>
> This is not a mistake but rather a point that could be clarified more explicitly. When we refer to $1024^{256}$, we are indicating the complexity of the data, not the complexity of the model parametrization. The quantity that you mentioned, $1024 \times 256$, corresponds instead to the complexity of the generated samples under $q_\theta(x_1 | x_{t_{n-1}})$.
>
> It is also worth noting that as the number of sampling steps increases, the complexity of the resulting composition of distributions also grows. As mentioned in our response to reviewer [McJt](https://openreview.net/forum?id=RBly0nOr2h&noteId=FJm17HbJsP) [Q. 1], this increase in sampling steps can lead to higher-quality samples and help mitigate issues related to factorization.
>
> **[R. 1] Fig 3's caption should clarify that images are generated in VQ-GAN latent spaces. I think it's boarderline misleading to not mention specifically in the caption that it's a GAN-based latent-space image experiments. GAN's latent spaces are not only of lower dimension but also much structural.**
>
> We will specify this aspect of training in the caption, as you have suggested.
>
> **Concluding remarks.** We hope that, with the above clarifications, you will kindly reevaluate our work and find it deserving of a higher rating.

---

### Official Review · Reviewer_atNH · 2025-03-14

**Overall Recommendation:** 4

**Summary:**

The paper addresses the Schrodinger Bridge (SB) problem for discrete spaces (categorical data). It proposes CSBM: a method that extends IMF (actually D-IMF) to discrete categorical spaces, proving theoretical convergence, propoing a concrete implementation and showing experimental evidence with two practical reference processes.  Evaluations on synthetic datasets, Colored MNIST, and CelebA demonstrate competitive or superior performance in unpaired image-to-image translation tasks compared to baseline methods (ASBM and DSBM).

## update after rebuttal
the authors addressed my concerns and i maintain my recommendation

**Claims And Evidence:**

(1) Theoretical Claim: The uniqueness and convergence of the discrete-time IMF procedure for categorical spaces.
Evidence: A formal theorem is provided, clearly proving the convergence under stated conditions. The proof appears rigorous to the best of my judgement
(2) practical algorithm: The proposed CSBM is claimed to be effective in practice for categorical SB problems.
Evidence:supported by experimental results on several datasets: gaussian to swiss roll, colored MNIST and VQ CelebA, demonstrating visually good translations and quantitative improvements  compared to baseline methods.

**Essential References Not Discussed:**

Hoogeboom et al., and Gat et al,  are mentioned but could use further discussion and perhaps comparison

**Experimental Designs Or Analyses:**

The experimental designs appear solid, with clear evaluation metrics (FID, CMMD) and visual evidence to assess qualitative performance. However, the chosen parameters for stochasticity level (alpha)  appear somewhat ad-hoc. further explanation of the choice of particular values would be helpful

**Methods And Evaluation Criteria:**

The methods and evaluation criteria (FID, CMMD) make sense given the problem of discrete unpaired translation. I liked the use of VQ representations of the celeb-A images as it aligns  with common practices in generative modeling. The select reference processes (Uniform and Gaussian-like) are well suited for common scenarios (unordered vs ordered categories).

**Other Comments Or Suggestions:**

none

**Other Strengths And Weaknesses:**

Strengths:

+ Clear motivation and justification for extending IMF to discrete spaces.
+ Thorough theoretical support and convincing experiments showing effectiveness across tasks and settings
+ Very clear presentation -- the paper is fun to read

Weaknesses:
- Could benefit from additional experiments on other categorical datasets (e.g., discrete text tokens or molecules) and perhaps more comparison with baselines.

**Questions For Authors:**

* Clarification: In Alg 1, when sampling (x0,x1) using q_eta or q_theta (x1|x0), do the authors mean x1 is simulated from x0 with N steps? or estimated directly from the x1 timestep? if it is the latter -- could you provide more details on how this is done in practice?

* Can the authors discuss ways to reduce the information loss arising from the factorization of the conditional distributions?

**Relation To Broader Scientific Literature:**

The authors do a really good job positioning their work within existing literature on SB, optimal transport, and diffusion models. They  distinguish their contributions from related methods -- continuous-space IMF (e.g., Shi et al., 2023; Gushchin et al., 2024) and the part clarifying difference from discrete optimal transport methods (Sinkhorn algorithm, gradient-based approaches) was also a good addition (even though i felt the distinction was clear, it was nice to read and to be stated explicitly).

**Theoretical Claims:**

Theorem 3.1's proof appears correct and rigorously executed

---

> ### Author Rebuttal · Authors · 2025-04-01
>
> Dear reviewer atNH thank you for your questions and commentaries.
>
> **[R. 1] Hoogeboom et al., and Gat et al, are mentioned but could use further discussion and perhaps comparison**
>
> Regarding the references [1, 2] you mentioned, we believe they are not suitable for comparison in our setting. For [1], the practical differences from D3PM [3] (which is our backbone method) are minimal, as [3] extends the discrete diffusion framework introduced in [1]. While the second method [2] could, in principle, be used as a backbone for our method, it is well known that lower values of $\epsilon$ (or $\alpha$ in our work) make it significantly harder for the model to approximate the target distribution (see [4], Figure 7). Current discussion will be added to the revised version.
>
> [1] Hoogeboom, Emiel, et al. "Argmax flows and multinomial diffusion: Learning categorical distributions." Advances in neural information processing systems 34 (2021): 12454-12465.
>
> [2] Gat, Itai, et al. "Discrete flow matching." Advances in Neural Information Processing Systems 37 (2024): 133345-133385.
>
> [3] Austin, Jacob, et al. "Structured denoising diffusion models in discrete state-spaces." Advances in neural information processing systems 34 (2021): 17981-17993.
>
> [4] Shi, Yuyang, et al. "Diffusion Schrödinger bridge matching." Advances in Neural Information Processing Systems 36 (2023): 62183-62223.
>
> **[W. 1] Could benefit from additional experiments on other categorical datasets (e.g., discrete text tokens or molecules) and perhaps more comparison with baselines.**
>
> We believe the current set of experiments is sufficient to support our claims. In particular, the practical implementation of the VQ-GAN setup does not differ significantly from potential setups with text data. The model and reference process will remain the same. While graphs are indeed more challenging due to their structural complexity, this domain has already been addressed by DDSBM. As explained in our response to reviewer [MBG2](https://openreview.net/forum?id=RBly0nOr2h&noteId=20eaTHc0ic) [W. 2], we do not include a direct comparison with DDSBM and instead focus on exploring other experimental settings.
>
> **[Q. 1] Clarification: In Alg 1, when sampling $(x_0,x_1)$ using $q_\eta$ or $q_\theta(x_1|x_0)$, do the authors mean $x_1$ is simulated from $x_0$ with $N$ steps? or estimated directly from the $x_1$ timestep? if it is the latter -- could you provide more details on how this is done in practice?**
>
> Your first suggestion is correct. To generate samples, we follow a standard diffusion sampling procedure. After the $l$-th step of D-IMF, we obtain $q^l_\theta(x_1 | x_{t_{n-1}})$. Moving to $(l+1)$-th D-IMF iteration, we first apply the trained model to generate $x_1$ taking as an input a dataset point $x_0$. We then perform posterior sampling using $q^{\text{ref}}(x_{t_n} | x_{t_{n-1}}, x_1)$ to obtain $x_{t_n}$. This procedure is repeated for $N+1$ steps to obtain samples from coupling $q^l_\theta(x_0, x_1)$. For the backward parametrization, we use the same scheme.
>
> **[Q. 2] Can the authors discuss ways to reduce the information loss arising from the factorization of the conditional distributions?**
>
> Please refer to our response to reviewer [McJt](https://openreview.net/forum?id=RBly0nOr2h&noteId=FJm17HbJsP) [Q. 1].
>
> **[R. 2] However, the chosen parameters for stochasticity level (alpha) appear somewhat ad-hoc. further explanation of the choice of particular values would be helpful**
>
> The pattern of selecting $\alpha$ follows the same intuition as choosing $\epsilon$ in continuous SB methods. Specifically, lower values of $\alpha$ lead to less stochasticity in the trajectories, resulting in higher similarity to the input data but a lower-quality approximation of the target distribution. At very low values, the model may collapse due to insufficient stochasticity. Conversely, higher values of $\alpha$ introduce more variability, improving the quality of the approximation but reducing similarity to the initial data. Beyond a certain point, excessively large $\alpha$ values make the model difficult to train, leading to a drop in both quality and consistency. Unfortunately, the effective range of these behaviors is highly dependent on the dataset and the chosen reference process. Nonetheless, we provide reasonable baseline values from which one can begin and adjust as needed.

---

### Official Review · Reviewer_MBG2 · 2025-03-19

**Overall Recommendation:** 2

**Summary:**

This paper introduces Categorical Schrödinger Bridge Matching, an approach that extends the Schrödinger Bridge (SB) framework to discrete spaces. While SB has gained traction in generative modeling and domain translation, most prior work has been confined to continuous spaces. The paper addresses this gap by developing a theoretical foundation and a computational algorithm tailored for discrete data.

## update after rebuttal

I have decided to maintain my score, as two main concerns remain:

1. It remains unclear how the discrete-time formulation offers meaningful insights to the field. While the authors argue that prior work relying on continuous-time models is theoretically unsound, I find that the improvements in mathematical rigor on such a fine-grained detail do not constitute a substantial breakthrough from a research perspective.  In particular, I do not perceive a theoretical barrier to translating the results from continuous to discrete time.

2. The similarity to DDSBM is also a concern, as the proposed framework appears to have significant overlap and does not clearly offer novel contributions beyond existing approaches.

**Claims And Evidence:**

**Limited Empirical Support for Discrete Claims:**

While the paper is motivated by discrete data applications (e.g., text, molecular graphs), all experiments focus on image-based tasks, which, despite using vector quantization, are inherently less discrete than the originally mentioned data types. A stronger demonstration on truly discrete datasets (e.g., molecular graphs, categorical sequences, or text-based data) would significantly bolster the claims.

**Essential References Not Discussed:**

**Missing Baseline: Discrete Static OT Solver:**

A missing baseline in this paper is a **discrete static optimal transport (OT) solver** built on top of **Reference [1]**. Since the proposed approach is fundamentally an extension of Schrödinger Bridge methods to discrete spaces, it is essential to test whether a **simpler discrete OT solver**—one leveraging existing **static OT formulations**—could serve as an effective alternative within the **bridge matching framework** along side ASBM and DSBM.



[1] Somnath et al., Aligned Diffusion Schrödinger Bridges, 2023.

**Experimental Designs Or Analyses:**

The experimental design does not seem to substantiate the advantages of the proposed algorithm; see my comments in **Methods And Evaluation Criteria**.

**Methods And Evaluation Criteria:**

A central issue with this paper is the lack of comprehensive comparison to existing methods. While the authors position CSBM as an advancement for discrete-state Schrödinger Bridge problems, several relevant baselines are missing from the experiments.

For example, DDSBM (Discrete Diffusion Schrödinger Bridge Matching) has been proposed specifically for graph-structured discrete data, yet it is not included in the evaluation. Given that DDSBM also addresses discrete Schrödinger Bridge problems, a direct comparison would be necessary to assess whether CSBM provides meaningful improvements or is simply an alternative formulation. The omission of such a baseline weakens the empirical claims of the paper. In particular, this paper would greatly benefit by replicating the experimental setup in the DDSBM paper and demonstrating improvement.

Additionally, while the paper compares CSBM to ASBM and DSBM, these are methods designed for continuous spaces, meaning they are not necessarily the most appropriate baselines for a method explicitly aimed at discrete settings. A fairer assessment would include methods developed for discrete generative modeling, such as categorical diffusion models.

**Other Comments Or Suggestions:**

N/A

**Other Strengths And Weaknesses:**

The paper is well-written and effectively conveys its ideas.

**Questions For Authors:**

- In what ways does the derived algorithm differ from DDSBM?

- What values does the discrete-time perspective bring to the SB framework for categorical data?

**Relation To Broader Scientific Literature:**

The problem studied in this paper is **important and valuable**, as extending **Schrödinger Bridge methods to discrete spaces** is highly relevant for applications like **molecular generation, text modeling, and vector-quantized representations**.

**Theoretical Claims:**

**Questionable Novelty of the Theoretical Contribution:**

The discrete-time theory presented in this paper appears to be a rather trivial extension of the existing continuous-time Schrödinger Bridge theory. The results largely follow from prior work and do not introduce fundamentally new insights beyond what has already been established in the continuous-time setting. While the authors claim to provide a theoretical foundation for discrete-time setting, it is unclear whether this contribution is substantially novel or merely a straightforward adaptation of known results.

Moreover, there is a deeper concern regarding the algorithm itself. The submission presents CSBM as a new computational approach, but it is unclear whether this is substantively different from DDSBM. DDSBM is explicitly motivated by continuous-time considerations, yet it is naturally discretized for implementation. Since the proposed CSBM method operates in discrete time, one must question: Is CSBM simply DDSBM with a different motivation? The paper does not make a clear distinction between the two, and without this clarification, the novelty of the algorithm is questionable.

If the authors aim to claim CSBM as a distinct method, they must explicitly differentiate it from DDSBM and explain what aspects of their approach do not follow directly from the standard discretization of the continuous-time Schrödinger Bridge framework.

---

> ### Author Rebuttal · Authors · 2025-04-01
>
> Dear reviewer MBG2, thank you for raising important questions regarding our paper.
>
> **[W. 1] Questionable Novelty of the Theoretical Contribution**
>
> We respectfully disagree. First, as highlighted in Table 1, the continuous-time frameworks DDSBM [1] and DSBM [2] rely heavily on the theoretical foundation established in [3], which does not extend to discrete time. Therefore, our work should not be viewed as a trivial extension of these frameworks but rather as a distinct theoretical foundation that shares certain similarities. Furthermore, the core theoretical contribution of our paper goes beyond simply generalizing the D-IMF procedure from [4] to discrete data: the discrete case emerges as a consequence of a broader generalization of the D-IMF procedure to **arbitrary reference processes**. As noted in the article (see footnote 1, page 5), our generalization enables ASBM [4] to operate with any Markov process $q^{ref}$.
>
> **[W. 2] Moreover, there is a deeper concern regarding the algorithm itself..., one must question: Is CSBM simply DDSBM with a different motivation?... For example, DDSBM has been proposed specifically for graph-structured discrete data, yet it is not included in the evaluation... In what ways does the derived algorithm differ from DDSBM?**
>
> We agree that adding practical distinctions will strengthen the comparison. Thus, we will include them in the revision.
>
> First, let us clarify how DDSBM derives its loss. In theory, it matches the generator matrices $A$ (see [1, Equations (5, 6)]). However, in practice, authors of [1] discretize time, which leads to minimization of $D_{KL}(q^{ref}(x_{t_n}|x_{t_{n-1}}, x_1)|| q_{\theta}(x_{t_n}|x_{t_{n-1}}))$ (see the derivation in [1, Appendix E.1]). Thus, as it could be mentioned, it is indeed the same loss as presented in our article. However, since we have theoretically alternative objective $D_{KL}(q^{ref}(x_{t_n}|x_{t_{n-1}})||q_{\theta}(x_{t_n}|x_{t_{n-1}}))$ (see Prop. 3.3), we can match distributions directly by employing **various loss functions** such as MSE or even adversarial training as in ASBM [4]. We conducted extra experiments using only an MSE loss and observed comparable to KL (**LINK**: see [figures](https://anonymous.4open.science/r/images-64B3) with prefix 'toy'). Thus, the similarity in practical implementation reflects our design choice to use this particular parametrization. Alternative approaches could be easily used. This also explains why we did not include a comparison with DDSBM.
>
> **[W. 3] A missing baseline discrete OT with [5]**
>
> If we correctly understand, you propose first using classical discrete OT methods to get aligned data followed by the application of [5]. However, there is one crucial problem with this setup. The SB problem that lies behind [5] is still considered to be continuous, i.e., continuous Brownian motion is used as a reference process. This breaks the assumptions on discrete data. We will cite [5] in the final revision and discuss it, but we think that a comparison with it is impossible due to the reasons mentioned above.
>
> **[W. 4] Additionally, while the paper compares CSBM to ASBM and DSBM, these are methods designed for continuous spaces, meaning they are not necessarily the most appropriate baselines...**
>
> To our knowledge, there are no other discrete domain translation models for unpaired data. Thus, the only methods we compare are continuous-space methods ASBM and DSBM.
>
> **[W. 5] ...all experiments focus on image-based tasks, which, despite using vector quantization, are inherently less discrete than the originally mentioned data types...**
>
> We apologize, but the meaning of "less discrete" is unclear to us. Still, for further clarification regarding the choice of experiments (especially texts), please refer to our response to the reviewer [atNH](https://openreview.net/forum?id=RBly0nOr2h&noteId=0v01XUlBGp) [W. 1].
>
> **[Q. 1] What values does the discrete-time perspective bring to the SB framework for categorical data?**
>
> The main theoretical advantage is the ability to consider CSBM with $N = 1$, which is guaranteed to converge to the SB. In contrast, continuous-time setups typically require to assume $N=\infty$ to achieve convergence. From a practical side, our framework also enables the flexible selection of loss functions for matching the transition distributions, which we mentioned in answers to your previous questions.
>
> **Concluding remarks.** We hope that, with the above clarifications, you will kindly reevaluate our work and find it deserving of a higher rating.
>
> [1] Kim, Jun Hyeong, et al. "Discrete Diffusion Schrödinger Bridge Matching for Graph Transformation."
>
> [2] Shi, Yuyang, et al. "Diffusion Schrödinger bridge matching."
>
> [3] Léonard, Christian. "A survey of the Schrödinger problem and some of its connections with optimal transport."
>
> [4] Gushchin, Nikita, et al. "Adversarial Schrödinger Bridge Matching."
>
> [5] Somnath, Vignesh Ram, et al. "Aligned diffusion schrödinger bridges."

---

### Official Review · Reviewer_McJt · 2025-03-23

**Overall Recommendation:** 4

**Summary:**

The paper does what the title says: it establishes the basic framework for the version of Schrodinger bridge diffusion models, for the case of discrete (categorical) spaces. This means that it has a theoretical result describing why and how an iterated projection method for finite-steps markov processes can be made to converge to the optimum bridge process, and then it describes how to implement this iteration in practice using neural networks, and shows a few examples to illustrate the phenomena.

**Claims And Evidence:**

Yes I think that the claims are all carefully argued and convincing.

**Essential References Not Discussed:**

I don't know of any.

**Experimental Designs Or Analyses:**

I did not check in detail the experiment implementation (i.e. I did not run the codes from the supplementary material), but I believe that the experiment outcomes are realistic and that the declared setup is sound.

**Methods And Evaluation Criteria:**

I think that the examples are toy model examples, even the VQ-VAE one which is the hardest is a toy model. I don't see a big issue with that, given that this is really the first formulation of SB's for categorical data, toy model experiments are to be awaited. Nevertheless, I may be biased towards theory, a referee interested in practical applications may require stronger evidence that the model helps to advance applications. (The fact that this model would be better than any competitor, is not part of the experimental validation.)

**Other Comments Or Suggestions:**

Here are a few places where some rewriting may help:

Line 110 column 2: "additional properties" sounds too vague, maybe state some examples

Line 235 column 2: "D=1" seems a weird way to put it, and it confused me because I had forgotten what D was.. maybe just say that you work with \mathbb S and it'll be clear that D=1

Line289 column 1: when you introduce this factorization, maybe briefly write about the limitation that this imposes (I know that this has been written in the "limitations" part, but I think it's worth putting it here too)

Line 290-299 column 1 : "Since, in fact, we need N+1 neural nets to do the prediction of endpoints at each time step, we simply use a single neural network with an extra input n" this sentence is unclear to me. I don't follow the implication beyond the "since [...]", and I don't see how "n" is an input of a neural network. I feel that this sentence tries to abbreviate stuff too much and it became unintelligible. Can you expand and state this clearly please?

Line 354-355 column 2: "discrete space of images but not continuous" this looks weird.. yeah it's discrete and not continuous, what did you want to say with the "but" part? Please correct/erase whatever is needed to make this right.

**Other Strengths And Weaknesses:**

A strength of the paper is its simplicity. Some referees may shun that, saying that the work is somehow minor since it is not technically involved, but I disagree.

**Questions For Authors:**

I'd like to know more precisely some examples of what kind of dependencies will be lost in the dimensional factorization, and also if the authors have any ideas on how to "do better than" the factorization S^D \to D\times S metioned in the un-numbered formula following (9). I mean, how would you increase a bit the complexity of the models to try to make it "forget less" about the dependencies between dimensions? In the case of a discrete space this is easier than in general, so it's worth giving some pointers.

**Relation To Broader Scientific Literature:**

As said before, the SB framework was previously restricted to continuous spaces, and no version for discrete spaces was available. Thus this paper fills a gap.

The gap was not hard to fill: it was sufficient to build similar versions of proofs as in cited reference [Guschin et al 2024b], but for the discrete space case, and no surprises appeared anywhere. Some simplifying tricks like the one for passing from S^D to S\times D for practical purposes, were inherited from previous literature.

Even if the goal of the paper does not face strong new difficulties, it is important that the gap in the literature has been filled. Also, the new theoretical result (grey box in the paper) is elegant and simple to state so it's worth publishing.

**Theoretical Claims:**

Yes, I checked all the proofs. Furthermore the main results are in line (similar hypotheses and theses) with well established versions as indicated from Table 1. This means that there is little surprise that the results are true, and on the positive side, it gives good reason to trust that the results are valid, even for people who would not have read the proofs.

---

> ### Author Rebuttal · Authors · 2025-04-01
>
> Dear Reviewer McJt, thank you for pointing out the unclear parts you encountered. We will revise the text where you have highlighted, as much as possible.
>
> **[R.1] Line 110 column 2: "additional properties" sounds too vague, maybe state some examples**
>
> In line 110, by "additional properties", we refer to the reciprocal and Markovian properties, which allow us to use diffusion models (specifically, bridge matching models) as the backbone of our domain translation framework. Without these properties, we would be limited to training one-step models such as GANs, which have since been largely replaced by modern diffusion-based approaches.
>
> **[R.2] Line 235 column 2: $D=1$ seems a weird way to put it, and it confused me because I had forgotten what $D$ was.. maybe just say that you work with $\mathbb{S}$ and it'll be clear that $D=1$**
>
> In line 235, the purpose of $D = 1$ is to simplify the transition matrices defined in Equations (7, 8). Since we are using factorization, we consider feature-wise rather than data-point-wise transition matrices. So, to maintain compactness of the text, we omit introducing $Q_n$ for arbitrary $D$, as it is not used in our approach, again, due to the factorization.
>
> **[R.3] Line289 column 1: when you introduce this factorization, maybe briefly write about the limitation that this imposes (I know that this has been written in the "limitations" part, but I think it's worth putting it here too)**
>
> We agree that it is indeed important to clarify the factorization in line 289. We will include a reference to the limitations section in order not to distract the reader from the flow of the main text.
>
> **[R.4] Line 290-299 column 1 : "Since, in fact, we need N+1 neural nets to do the prediction of endpoints at each time step, we simply use a single neural network with an extra input n" this sentence is unclear to me...**
>
> In lines 290–299, we intended to explain that we use a single neural network for all time steps rather than separate networks for each step, which is a common practice. To simulate the stochastic process, one would typically require $N+1$ distinct functions $q(x_1 | x_{t_{n-1}})$ for each $n$. However, training $N+1$ neural networks is computationally expensive. Instead, we use a single neural network for all transition steps (i.e., the transition function $q(x_1 | x_{t_{n-1}})$), with additional time conditioning $q_\theta(x_1 | x_{t_{n-1}}, t_{n-1})$ for all $n \in [1, N+1]$. If you are also interested in the sampling procedure using the trained model, please refer to our response to the reviewer [atNH](https://openreview.net/forum?id=RBly0nOr2h&noteId=0v01XUlBGp) [Q. 1].
>
> **[R.5] Line 354-355 column 2: "discrete space of images but not continuous" this looks weird.. yeah it's discrete and not continuous, what did you want to say with the "but" part? Please correct/erase whatever is needed to make this right.**
>
> In lines 354–355, we intended to highlight that image spaces are typically treated as continuous rather than discrete, i.e., the space of image pixels is commonly represented as the interval $[0, 1]^D$ rather than the discrete set {$0, 1, \dots, 255^D$}.
>
> **[Q. 1] I'd like to know more precisely some examples of what kind of dependencies will be lost in the dimensional factorization, and also if the authors have any ideas on how to "do better than" the factorization $S^D \to D\times S$ metioned in the un-numbered formula following (9). I mean, how would you increase a bit the complexity of the models to try to make it "forget less" about the dependencies between dimensions? In the case of a discrete space this is easier than in general, so it's worth giving some pointers.**
>
> Regarding factorization, to the best of our knowledge, the work [1] is the only existing work addressing this issue. Authors introduce an additional generative model that models the copula of the factorized distributions, which is, informally, the part of the joint distribution $q_{\theta}(x_1 | x_{t_{n-1}}) = q_{\theta}(x^1_1, ..., x^D_1 | x_{t_{n-1}})$ that is lost due to the factorization.
>
> However, practically, it seems that the issue of factorizayion can also be mitigated just by increasing the number of steps, as demonstrated, for example, in our experiments with C-MNIST. The more steps that are taken during sampling, the more expressive the resulting composition of distributions becomes. As a result, repeatedly incorporating the full previous state leads to more correlated features. Even though factorization has implications for our practical implementation, it is important to recall, as stated in the article, that this issue is inherent to all discrete diffusion models.
>
> [1] Liu, Anji, et al. "Discrete Copula Diffusion." arXiv preprint arXiv:2410.01949 (2024).

---

### Decision · Program_Chairs · 2025-05-01

**Decision:**

Accept (poster)

**Comment:**

This paper extends the Schrödinger Bridge (SB) framework to discrete spaces by developing a theoretically grounded and practically implementable algorithm based on Iterative Markovian Fitting (IMF). The proposed method is validated through rigorous theoretical analysis and empirical studies on synthetic data and quantized image representations.
While some reviewers expressed concerns regarding novelty and similarity to recent works, such as DDSBM, the authors provided clear justifications, including theoretical distinctions and flexible algorithmic design. In contrast, several reviewers found the contribution meaningful, particularly for filling an important gap in the literature and providing a solid foundation for discrete SB. The experiments, while limited to image data, are carefully conducted and demonstrate competitive performance.
Given that some negative comments remain, for example, especially on empirical scope and relation to prior work, the paper makes a significant contribution to the field by formalizing the discrete Schrödinger Bridge problem and providing a viable algorithmic solution. Therefore, I recommend acceptance.